# Peroxy acetyl nitrate (PAN) measurements at northern mid-latitude mountain sites in April: A constraint on continental source-receptor relationships

Arlene M. Fiore[1,2], Emily V. Fischer[3], George P. Milly[2], Shubha Pandey Deolal[4], Oliver Wild[5], Dan Jaffe[6,7], Johannes Staehelin[4], Olivia E. Clifton[1,2], Dan Bergmann[8], William Collins[9], Frank Dentener[10], Ruth M. Doherty[11], Bryan N. Duncan[12], Bernd Fischer[13], Stefan Gilge[14,15], Peter G. Hess[16], Larry W. Horowitz[17], Alexandru Lupu[18,19], Ian MacKenzie[11], , Rokjin Park[20], Ludwig Ries[21], Michael G. Sanderson[22], Martin G. Schultz[23], Drew T. Shindell[24], Martin Steinbacher[25], David S. Stevenson[11], Sophie Szopa[26], Christoph Zellweger[25], Guang Zeng[27]

[1]Department of Earth and Environmental Science, Columbia University, Palisades, NY, 10964, U.S.A.
[2]Lamont-Doherty Earth Observatory of Columbia University, Palisades, NY, 10964, U.S.A.
[3]Department of Atmospheric Science, Colorado State University, Fort Collins, CO, 80521, U.S.A.
[4]Institute for Atmospheric and Climate Science, ETH Zürich, Switzerland
[5]Lancaster Environment Centre, Lancaster University, Lancaster, LA1 4YQ, UK
[6]School of STEM, University of Washington, Bothell, WA, 98011, U.S.A.
[7]Department of Atmospheric Science, University of Washington, Seattle, WA, 98195, U.S.A.
[8]Lawrence Livermore National Laboratory, Livermore, CA, 94550, U.S.A.
[9]Department of Meteorology, University of Reading, Reading, RG6 6BB, UK
[10]European Commission, Joint Research Centre, Ispra, I-21027, Italy
[11]School of GeoSciences,,The University of Edinburgh, Edinburgh,  EH9 3FF, UK
[12]Atmospheric Chemistry and Dynamics Laboratory, NASA GSFC, Greenbelt, MD 20720, U.S.A
[13]Federal Environment Agency (UBA), Schauinsland, 79254, Germany
[14]Meteorological Observatory Hohenpeissenberg, German Meteorological Service (DWD), Hohenpeissenberg, DE
[15]now at DWD, Research Center Human Biometeorology, Freiburg, DE
[16]Department of Biological and Environmental Engineering, Cornell University, Ithaca, NY, 14853, U.S.A.
[17]Geophysical Fluid Dynamics Laboratory, National Oceanic and Atmospheric Administration, Princeton, NJ, 08540, U.S.A.
[18]Centre for Research in Earth and Space Science, York University, Toronto, M3J 1P3, Canada
[19]now at Air Quality Research Division, Environment and Climate Change Canada, Toronto, M3H 5T4, Canada
[20]School of Earth and Environmental Sciences, Seoul National University, Seoul, 08826, Republic of Korea
[21]II4.5.7,German Environment Agency (UBA), Zugspitze, 82475, Germany
[22]Met Office, Exeter, EX1 3PB, UK.
[23]Jülich Supercomputing Centre, Forschungszentrum Jülich, 52425 Jülich, Germany
[24]Nicholas School of the Environment, Duke University, Durham, NC, 27708, U.S.A.
[25]Laboratory for Air Pollution / Environmental Technology, Empa – Swiss Federal Laboratories for Materials Science and Technology, Dübendorf, CH-8600, Switzerland
[26]Laboratoire des Sciences du Climat et de l'Environnement, Institut Pierre Simon Laplace, CEA/CNRS/UVSQ, Gif sur Yvette, France
[27]National Institute of Water and Atmospheric Research, Wellington, 6021, New Zealand

*Correspondence to*: Arlene M. Fiore (amfiore@ldeo.columbia.edu)

**Abstract.** Abundance-based model evaluations with observations provide critical tests for the simulated mean state in models of intercontinental pollution transport, and under certain conditions may also offer

constraints on model responses to emission changes. We compile multi-year measurements of peroxy
acetyl nitrate (PAN) available from five mountaintop sites and apply them in a proof of concept approach
that exploits an ensemble of global chemical transport models (HTAP1) to identify an observational
"emergent constraint". In April, when the signal from anthropogenic emissions on PAN is strongest,
simulated PAN at northern mid-latitude mountaintops correlates strongly with PAN source-receptor
relationships (the response to 20% reductions in precursor emissions within northern mid-latitude
continents; hereafter, SRRs). This finding implies that PAN measurements can provide constraints on PAN
SRRs by limiting the SRR range to that spanned by the subset of models simulating PAN within the
observed range. In some cases, regional anthropogenic volatile organic compound (AVOC) emissions,
tracers of transport from different source regions, and SRRs for ozone also correlate with PAN SRRs.
Given the large observed interannual variability in the limited available datasets, establishing strong
constraints will require matching meteorology in the models to the PAN measurements. Application of this
evaluation approach to the chemistry-climate models used to project changes in atmospheric composition
will require routine, long-term mountaintop PAN measurements to discern both the climatological SRR
signal and its inter-annual variability.
**1 Introduction**
Peroxy acetyl nitrate (PAN) is produced alongside ozone ($O_3$) from photochemical reactions involving
precursor emissions of nitrogen oxides ($NO_x$) and non-methane volatile organic compounds (VOC). Once
ventilated from a source region to the free troposphere where it is more stable at colder temperatures, PAN
can be efficiently transported throughout the hemisphere (Singh, 1987; Singh and Hanst, 1981). When a
PAN-containing free tropospheric air mass subsides, PAN thermally decomposes to release $NO_x$ and can
thus facilitate $O_3$ formation far downwind (Wild et al., 1996; Schultz et al., 1999; Jaeglé et al., 2003;
Kotchenruther et al., 2001a; Hudman et al., 2004). Both PAN and $O_3$ distributions over any northern mid-
latitude region reflect the combined influence of production from sources within the region and transport
from outside that region. At northern mid-latitudes, the intercontinental influence from anthropogenic
emissions on surface $O_3$ levels is largest during spring (e.g., HTAP 2010) and occurs via at least two
pathways: (1) $O_3$ can be produced within a polluted continental boundary layer, ventilated to the free
troposphere and efficiently transported to other continents; and (2) $O_3$ can be produced in transit from the
export and subsequent chemical evolution of PAN and other precursors. Below, we examine the extent to
which springtime PAN observations at northern mid-latitude mountaintop sites can be used to constrain the
spread in multi-model estimates of source-receptor relationships (SRRs), where the sources are continental-
scale regions and the receptors are the mountaintop sites, for both PAN and $O_3$.
Observations during several aircraft field campaigns in the Eastern Pacific and at mountain top sites in the
Western U.S. and North Atlantic document efficient $O_3$ production in the lower troposphere following
subsidence of PAN-containing air masses (Fischer et al., 2010; Heald et al., 2003; Hudman et al., 2004;
Kotchenruther et al., 2001a,b; Val Martin et al., 2008; Zhang et al., 2008). When PAN decomposes in low-
$NO_x$ regions of the atmosphere, the $NO_x$ released can produce $O_3$ up to eight times more efficiently than in
polluted (high-$NO_x$) regions (Liang et al., 1998; Liu et al., 1987) and thus increase global $O_3$ abundances
(Moxim et al., 1996; Wang and Jacob, 1998), as $O_3$ formation is $NO_x$-limited in most of the free
troposphere (Chameides et al., 1992). The lifetime of PAN against thermal decomposition is about 1 hour
at 20°C, and it approximately doubles for every 4°C decrease in temperature, leading to a lifetime of at
least a month in the mid-troposphere during spring. This strong temperature dependence implies that a
warmer climate will decrease PAN export from polluted continental boundary layers, although a rise in
temperature-sensitive biogenic precursor emissions may temper this response (e.g. Doherty et al., 2013).
Future projections of atmospheric composition under global change scenarios will thus benefit from a
thorough understanding of the role PAN plays in transporting oxidized reactive nitrogen and thereby
altering ozone production throughout the troposphere.

To better distinguish among disparate estimates for intercontinental $O_3$ transport in the published literature,
the Task Force on Hemispheric Transport of Air Pollution (HTAP) organized an international global
modelling study, referred to here as HTAP1.  The HTAP1 study identified a factor of two range across
individual model estimates of surface $O_3$ response to changes in anthropogenic precursor emissions from
continental-scale, northern mid-latitude source regions (HTAP 2007; Fiore et al., 2009; HTAP 2010; Wild
et al. 2012).  The HTAP1 models do not distinguish between intercontinental $O_3$ transport occurring due to
$O_3$ produced from PAN chemistry versus direct transport of $O_3$ formed in a remote boundary layer, but
other work indicates that both pathways contribute.  Jaegle et al. (2003) find that 28% of the $O_3$ in the
Pacific Northwest free troposphere between 0-6 km is associated with PAN-to-$NO_x$ conversion, consistent
with Jiang et al. (2016) who found that  PAN produced from East Asian emissions and exported to the free
troposphere contributes 35% and 25% in spring and summer, respectively, to the free tropospheric $O_3$
abundance over western North America.  Over East Asia, Lin et al. (2010) found that the export of PAN
produced from European anthropogenic emission changes and subsequent downwind $O_3$ formation
contributed 20% of the spatially averaged response of surface $O_3$ levels, and up to 50% of the $O_3$ response
at mountain sites.

In addition to the direct influence of PAN on intercontinental $O_3$ transport, PAN may serve as a sensitive
diagnostic of model uncertainties in $O_3$ production chemistry and transport (Emmerson and Evans, 2009;
Kuhn et al., 1998). Prior analysis of measurements and global model simulations suggests that PAN
abundances at high altitude sites may be more sensitive than $O_3$ itself to changes in precursor emissions
(Fiore et al., 2011; Fischer et al., 2011; Jaffe et al., 2007).  We interpret this stronger sensitivity of PAN
than $O_3$ to changes in precursor emissions as reflecting buffering of $O_3$ by compensating changes to $O_3$
losses, whereas PAN loss pathways are far less sensitive to changes in precursor emissions. PAN loss
pathways include thermal decomposition (which dominates below approximately 7 km); photolysis in the
upper troposphere; and dry deposition within the boundary layer (Kirchner et al., 1999; Roberts, 2007;
Turnipseed et al., 2006). All of the HTAP1 models include PAN formation, but the chemical mechanisms
and kinetic rate coefficients differ, with likely implications for long-range transport (Emmerson and Evans,
2009; Knote et al., 2015). A prior multi-model study found that even with the same emissions, PAN differs
widely across models, reflecting differences in simulated photochemistry (Emmons et al., 2015). While the
absence of direct emissions and its low background make PAN a useful tracer of photochemistry, we note
that $O_3$ typically responds more strongly to changes in $NO_x$ emissions, while PAN responds more strongly
to changes in VOC emissions in many regions (Fischer et al., 2014; see their Figure 4).

A challenge in discriminating among model estimates of $O_3$ produced from different source regions is the
lack of direct observational constraints on SRRs. For example, Fiore et al. (2009) did not find any
relationship across models between their biases against surface $O_3$ observations and the strength of their
response to emission changes. In the absence of an observable quantity to constrain these relationships,
one approach is to identify an "emergent constraint" (Borodina et al., 2017), whereby a non-observable
quantity correlates strongly across a multi-model ensemble with an observed variable. The inter-model
range of the non-observable quantity is then narrowed by limiting it to the range encompassed by the
models closest to the observed variable. This approach has gained traction for narrowing the spread across
future climate projections (e.g., Hall and Qu, 2006; Cox et al. 2018). In light of its role as a proxy for ozone
formation chemistry, its direct role in facilitating intercontinental ozone transport, and the large signature of
PAN originating in the European boundary layer during spring found at Jungfraujoch (Pandey Deolal et al.,
2013; 2014), we hypothesize that PAN measurements may offer much-needed constraints for
discriminating across model estimates of intercontinental transport of PAN, and possibly $O_3$. The number
of models contributing to the HTAP1 study, which was designed to maximize comparability across
individual model estimates of ozone responses to changes in precursor emissions within northern mid-
latitude continental-scale source regions, offers an opportunity to evaluate this hypothesis.

We describe the HTAP1 model simulations, mountaintop measurements and our strategy to sample the
models at these sites (Section 2) before illustrating our rationale for selecting the month of April to quantify
the range of multi-model PAN distributions and PAN measurements at northern mid-latitude mountain sites
(Section 3). We then borrow from the "emergent constraint" approach in climate science to show that
correlations between simulated total PAN and SRRs for PAN are sufficiently strong as to permit PAN
measurements at mountaintop sites (one in each of the three major mid-latitude source regions) to narrow
the wide inter-model spread in estimates of PAN origin (Section 4). We further examine inter-model
relationships between the simulated PAN SRRs at these three mountaintop sites and regional precursor
emissions, and with a proxy for model transport (Section 5). Finally, we assess the relationship between
PAN and $O_3$ SRRs (Section 6) and conclude with a summary and recommendations for future work based
on our proof of concept analysis (Section 7).

   **2. Approach**

   **2.1 HTAP1 model simulations**

We use monthly mean PAN mixing ratios for the year 2001 simulated by fourteen global chemistry
transport models (Table 1); the temporal resolution for three-dimensional chemical fields archived from the
HTAP1 models is limited to monthly. We use four HTAP1 Source-Receptor (SR) simulations (Table 2): a
base case (SR1) and three perturbation simulations in which anthropogenic $O_3$ precursor emissions ($NO_x$,
VOC, carbon monoxide and aerosols) are reduced simultaneously by 20% within East Asia (SR6EA),
Europe and northern Africa (SR6EU), and North America (SR6NA). We calculate PAN SRRs by
differencing the perturbation and base simulations (SR1-SR6XX), where XX refers to the region in which
emissions of PAN precursors were decreased by 20%.

Of the models in Table 1, eleven used 2001 meteorological fields. Two models are chemistry-transport
models coupled directly to a general circulation model forced by observed sea surface temperatures
(STOC-HadAM3 and STOCHEM) and one model incorporates chemistry directly into a general circulation
model (UM-CAM). We include these models as our evaluation compiles PAN measurements across
several years (Section 2.2). The individual model specifications and emissions are described in Tables 1
and 2 of Fiore et al. (2009). For HTAP1, each model used its own emissions inventories (see Table A1 of
Fiore et al., 2009); Fiore et al. (2009) provide emission totals within each HTAP1 source region for all
(their Table A2) and anthropogenic (their Table A3) emissions of $NO_x$, NMVOC, and CO. The relative
inter-model spread in regional anthropogenic emissions is smallest for $NO_x$ emissions in EU and NA
(<10%) and largest for VOC from EU (58%) (Fiore et al. 2009).

To separate the role of inter-model differences in transport from the combined impacts of inter-model
differences in emissions and chemistry on simulated PAN at the mountaintop sites, we analyze an
additional set of idealized tracer simulations available from eleven models (COfromXX in Table 1, where
XX is the source region). In these simulations, a set of tagged carbon monoxide-like tracers are emitted,
each from a single HTAP1 source region with a 50-day lifetime, and with identical emissions across
models. Biomass burning emissions for the CO tracers are from GFED (van der Werf et al., 2006; 2010)
and other emissions are from the RETRO project (Schultz et al., 2007; 2008). We refer to these tracers as
"COfromEA", "COfromNA", and "COfromEU", which denote the tracers emitted from EA, NA, and EU,
respectively (Table 2; see also Doherty et al., 2013 and Shindell et al., 2008).
**2.2 Multi-year PAN measurements at mountaintop sites and model sampling**
To evaluate the HTAP1 models, we compiled April mean climatologies of lower tropospheric PAN
measurements from northern mid-latitude mountain observatories (Table 3). Given the large interannual
variability in PAN abundances, we require at least two years of observations in April. PAN observations
from Mount Bachelor (U.S.A.), Jungfraujoch (Switzerland), and Zugspitze (Schneefernerhaus),
Hohenpeissenberg, and Schauinsland (all in Germany) meet these criteria. Taken together, these
mountaintop measurements span 15 years, from 1995 to 2010 (Table 3), although only one site
(Schauinsland) overlaps with the HTAP1 simulation year of 2001.
PAN was measured at all five mountain sites using gas chromatography with electron capture detection
(ECD). A custom system using a Shimazu Mini-2 ECD was employed at Mount Bachelor (Fischer et al.,
2010). The commercially available Meteorologie Consult (GmbH) system was used at the European sites
(Zellweger et al., 2000). Calibrations generate PAN from the photolysis of excess acetone and NO in air
(Warneck and Zerbach, 1992; Volz-Thomas et al., 2002). Reported detection limits are ~20 ppt for PAN
measurements at Mount Bachelor, and ~50 ppt for the European sites, with total uncertainties of <10%
(Fischer et al., 2010; Zellweger et al., 2003).
We include all available data at these sites without filtering for upslope winds or any other criteria. At
Mount Bachelor, the cleanest of the 5 mountaintop sites (Supplemental Figure 1), Fischer et al. (2010) have
shown that PAN mixing ratios are not primarily controlled by diurnal wind patterns, which lead to
variations an order of magnitude smaller than the total observed range in measured PAN. When
measurements fall below the detection limit, we include half of the detection limit. This assumption should
not affect our conclusions as mountaintop sites generally sample free tropospheric air at night (e.g., Weiss-
Penzias et al., 2004) but PAN values below the detection limit typically occur due to deposition in a
shallow nocturnal boundary layer.
For comparison with the observations, we sample each model on its native grid (Table 1) at the horizontal
grid cell containing the latitude and longitude of each mountain site. Orography at these mountain sites is
poorly resolved at the relatively coarse HTAP1 model horizontal resolutions. This mismatch requires us to
apply some approximations for vertical sampling. We convert the station altitude to an approximate
pressure level by assuming a mean tropospheric temperature of 260 K, and a corresponding atmospheric
scale height of 7.6 km. We then use monthly mean pressure fields from each model to linearly interpolate
PAN based on the pressures of the two model grid cells that vertically bound the station pressure. While
different sampling strategies may alter the exact value of simulated PAN and its comparison to
observations, our primary interest is in the inter-model differences. Although the Zugspitze and
Hohenpeissenberg sites fall within the same horizontal grid cell in the HTAP1 models, the station altitudes
differ, so we consider the two sites separately.
Given that we seek constraints on intercontinental transport from the three major mid-latitude source
regions, we conduct a more in-depth analysis at the highest altitude European site (Jungfraujoch), the most
likely of the available sites to measure PAN transported between continents in the free troposphere, as well
as at Mount Bachelor in North America.  At Jungfraujoch, we also evaluate SRRs in the models with an
estimate of PAN originating in the European boundary layer based on an analysis of 20-day back
trajectories (Pandey Deolal et al., 2013).  We conduct a proof of concept analysis at Mount Waliguan in
Asia (36.28°N, 100.90°E, 3816 m) to assess the potential for future PAN measurements at this site to
narrow the inter-model range in SRRs. Short-term measurements have previously been collected at this site
(Xue et al., 2011). While aircraft and satellite observations have advanced the understanding of the
chemistry and dynamics of individual PAN plumes using models that archived higher temporal frequency
chemical fields (e.g., Alvarado et al., 2010; Payne et al., 2014; Emmons et al., 2015), their limited temporal
coverage is not well suited for comparison with the HTAP1 monthly mean PAN mixing ratios.
**3. Modeled and measured lower tropospheric PAN at northern mid-latitudes in April**
Our goal is to assess the potential for mountaintop PAN measurements to discriminate among model
estimates of PAN and $O_3$ produced by regional anthropogenic emissions and transported to the
mountaintop sites. We thus focus our analysis on April when measured PAN reaches its seasonal maximum
(Penkett and Brice, 1986; Singh and Salas, 1989; Bottenheim et al., 1994; Schmitt and Volz-Thomas, 2004;
Supplemental Figure 1) and when the HTAP1 models indicate that the production of PAN from the EA,
EU, and NA source regions dominates total simulated PAN (Figure 1).  April thus offers the strongest
possible signal of the influence of anthropogenic emissions from these three northern mid-latitude source
regions in the mountaintop measurements.

Figure 2 shows the spatial distribution of the HTAP1 model ensemble mean PAN mixing ratios at 650 hPa
(~3 km), the level sampled by the highest altitude sites on which we focus the majority of our analysis.
PAN mixing ratios in April generally increase with latitude, as expected from the strong thermal
dependence of the PAN lifetime, although some of the highest mixing ratios are simulated over the Asian
source region. The multi-model spread in lower tropospheric PAN, represented by the coefficient of
variation (standard deviation over the 14 models divided by the model ensemble mean) is within ±45%
across much of the northern hemisphere (Figure 2).  The large inter-model spread over much of Europe in
Figure 2b implies that observational constraints in this region would be particularly valuable.

Observed and modelled PAN mixing ratios at the northern mid-latitude mountain sites are compared in
Figure 3; see Supplemental Figure 1 for a comparison extended throughout the year).  We consider the
measured range across years to bound the "plausible" portion of the wide range in simulated total PAN
across the models.  The multi-model mean falls in the range of the measurements at four of the sites, but is
higher than observed in any year at Mount Bachelor.  The model rankings show some consistency across
the different sites, suggesting systematic model differences that can be narrowed with a limited set of
observational constraints, especially for models that rank similarly across the sites on all three continents
(Figure 3). For example, CAMCHEM and GEMAQ are consistently at the higher end of the range while
GISS-PUCCINI and LLNL-IMPACT are at the low end. The two models falling closest to the observed
2001 value at Schauinsland (MOZECH and MOZARTGFDL) fall into the observed range at either Mount
Bachelor or Jungfraujoch; we analyze these two sites further in the following sections.

The longest observational dataset at Schauinsland varies by over a factor of three across years, consistent
with large inter-annual variability found in prior analyses at mountaintop sites (Zellweger et al., 2003;
Fischer et al., 2011; Pandey Deolal et al. 2013; 2014). All but one of the models (LLNL-IMPACT) fall
within the wide range of observed inter-annual variability at Schauinsland, underscoring the tenuous nature
of conclusions regarding model performance drawn from short observational records unless the modelled
and observed meteorological years match. Future work to coordinate consistent time periods between
measurements and models would provide tighter constraints than are possible with our proof-of-concept
analysis described in the following sections.
**4. Exploring emergent constraints on model SRRs from measured total PAN**
The range of the PAN SRRs across the HTAP1 models at Jungfraujoch, Mount Bachelor, and Mount
Waliguan is wide for all three source regions, spanning a factor of five or more in several cases (Figure 4).
The key to a successful emergent constraint analysis is for this range in inter-model PAN SRRs, our
unobservable quantity, to correlate with the total PAN simulated at the mountaintop site, our observable
variable. The strongest correlations emerge for PAN originating in the region where the mountain is
located, but some intercontinental SRR pairs also show significant correlations ($p \leq 0.05$) with total
simulated PAN (Figure 4).

We illustrate here how PAN measurements can be used to narrow the inter-model range in the SRR pairs.
For the sites with significant correlations, the range across years (i.e., red vertical lines in Figure 4) bound
the April mean values observed at Jungfraujoch and Mount Bachelor. The models falling in this range are
highlighted in red. We select these models to narrow the range in SRRs, indicated by the red horizontal
dashed lines extending from the bounding models (red symbols) to the ordinate axis. Figure 4 shows that
the constraint from total measured PAN narrows the inter-model range in SRRs for PAN by at least half,
revealing some models as outliers. Other models simulate SRRs within the observationally constrained
range (between the dashed red horizontal lines) despite falling outside the observed range for total PAN,
possibly indicating a role for inter-model differences in non-anthropogenic sources of PAN or in the
relative contributions from the individual mid-latitude source regions, which we investigate further in the
next section. Given the year-to-year variability in total PAN, stronger constraints could be placed in future
work where the model meteorology corresponds to the same year as the measurements.

At Jungfraujoch, we additionally consider PAN SRRs for the EU source region with those estimated previously by back-trajectory analysis (Pandey Deolal et al., 2013). While Pandey Deolal et al. (2013) also attribute trajectories to NA and EA, fewer than 15% and 4% of trajectories are attributed to those regions as compared to 25-50% from EU (range across years; see Figure 1 of Pandey Deolal et al. (2013)). Combining these low frequencies with the inevitable growth in uncertainty as trajectories lengthen, we have the most confidence in the Pandey Deolal et al. (2013) estimates for the EU region. The horizontal blue dashed lines indicate the bounds obtained from this trajectory-based approach to estimating PAN from EU. The models falling in these bounds overlap with those constrained by the total PAN measurements, lending some confidence that these two independent approaches (one using total PAN and the correlated inter-model spread in SRRs; the other using back-trajectories to estimate SRRs) yield useful constraints on the influence of the EU source region on PAN measured at Jungfraujoch.

We note that for consistency with the model SRRs in Figure 4, which are the responses to 20% emission reductions in the source region, we divide the Pandey Deolal et al. (2013) EU SRRs by five to scale back from their estimated "full contribution" (100%). This linear scaling of the PAN response between 20% and 100% may incur errors due to non-linear chemistry. With an additional simulation in which the FRSGCUCI model sets European anthropogenic emissions of $NO_x$, CO and VOC to zero (a 100% perturbation), we estimate this error to be ~10%. For intercontinental regions, this error reduces to < 3%. Earlier work shows that the smaller non-linearity in PAN for intercontinental versus regional source-receptor pairs also holds for ozone (Fiore et al., 2009; Wu et al., 2009; Wild et al., 2012), and demonstrates approximate linearity between the simulated tropospheric ozone burden and ±50% of present-day global $NO_x$ emissions (Stevenson et al., 2006).

**5. Factors contributing to the inter-model range in PAN SRRs**

We investigate the role of inter-model differences in regional emissions of PAN precursors versus transport in contributing to inter-model differences in the PAN response to continental-scale emission changes at the three mountaintop sites shown in Figure 4. At each site, we examine the correlation across models between simulated PAN SRRs and regional anthropogenic emissions of VOC (AVOC; Figure 5) or $NO_x$ (ANO$_x$). The relationships for the EA SRRs are not significant, even at Mount Waliguan. We find, however, that the inter-model range in regional AVOC emissions explains as much as 64% of the variation in PAN attributed to EU emissions, and at least 25% of the variance in PAN attributed to the NA region (Figure 5). In contrast to AVOC, we find little relationship between the range in simulated PAN SRRs at the mountain sites and the model spread in regional ANO$_x$ emissions. Fischer et al. (2014) have previously shown that PAN abundances respond more strongly to changes in emissions of VOC than of $NO_x$. Our analysis supports that earlier finding and furthermore highlights a key role for model differences in regional AVOC emissions in contributing to the inter-model range in PAN SRRs.

Differences in model transport (e.g., Arnold et al., 2015; Orbe et al., 2017) may also contribute to the inter-
model differences in PAN SRRs. Our analysis of the HTAP1 idealized CO tracers, however, reveals little
correlation between inter-model differences in these idealized tracers (which have identical regional
emissions and lifetimes applied in all of the models) and in the PAN SRRs sampled at these sites. Although
we do not find any clear overall correlation, differences in the idealized CO tracers explain some of the
scatter in Figure 5. For example, at Jungfraujoch for EU AVOC emissions of 22 Tg C a$^{-1}$, the lowest model
(GISS-PUCCINI) has one of the smallest values for the COfromEU tracer, whereas the highest model
(STOC-HadAM3) has the largest value of COfromEU.

In light of the dependence of inter-model differences in PAN attributed to EU and NA during April and the
corresponding regional AVOC emissions, we illustrate how one could extend our emergent constraints in
Figure 4 (horizontal dashed red lines) to the regional AVOC emission estimates shown in Figure 5. A
major caveat underlying this analysis is the mis-match between meteorological years for the models and
measurements as discussed above, and the underlying assumption that the relationships in Figure 5 can
exclusively be attributed to differences in the AVOC emissions (as opposed to chemistry or transport).
The observationally-constrained SRRs between PAN from NA and total PAN measured at Jungfraujoch
and Mount Bachelor can be used to narrow the range of NA AVOC emissions to 12-18 Tg C a$^{-1}$ (the low
end is ruled out by the constraint imposed by PAN from NA at Jungfraujoch; the high end is ruled out by
PAN from NA at Mount Bachelor). Similarly, the range for EU AVOC emissions would narrow to 16-25
Tg C a$^{-1}$.

We consider next the importance that various models ascribe to a given source region relative to another
source region. We first correlate the ratios of PAN from two different source regions with the total PAN
simulated by the individual models in April. We find little relationship, with the exception of Mount
Bachelor, where the observational constraint implies that more PAN originating from EA should be present
at Mount Bachelor than PAN originating from NA (Figure 6a). We interpret this as indicating that models
with higher total PAN at Mount Bachelor are overestimating North American influence at this mountain
site (which samples free tropospheric air). This interpretation is supported by the idealized CO tracer
simulations (with identical regional emissions and the same lifetime applied in all the models), which
suggest that some of the variance in the ratio of PAN from NA versus EA at Mount Bachelor is due to
differences in transport from the two regions (Figure 6b). We emphasize that these transport differences do
not simply reflect the use of different meteorology to drive the CTMs (Figure 6b).

By comparing NA:EA at Mount Bachelor, EU:NA at Jungfraujoch, and EA:EU at Mount Waliguan, we
examine the relative importance of emissions within the source region, where the measurement site is
located, versus the upwind intercontinental source region on PAN (Supplemental Figure 2). At Mount
Bachelor, the HTAP1 multi-model mean SRRs from NA, EA, and EU, are roughly equal in April (Figure
1). The differences across the HTAP models in the relative importance of the NA:EA source regions on
PAN (which range from about 0.5 to 2.5) correlate roughly equally with the ratio of the NA:EA CO
transport tracers and with the ratio of the NA:EA AVOC emissions (spearman rank correlation coefficient
(r) = 0.6 for both cases); we find no relationship with the ratio of the NA:EA $ANO_x$ emissions (Figure 6b
and Supplemental Figure 2 left column). At Jungfraujoch, the HTAP1 multi-model mean attributes much of
the PAN to emissions from the EU and NA source regions during April (Figure 1). The ratio of PAN
attributed to EU versus NA at Jungfraujoch, however, varies from approximately 0.5 to 2 across the
individual HTAP1 models (Supplemental Figure 2). In contrast to our findings at Mount Bachelor, this
ratio at Jungfraujoch depends most strongly on the ratio of $ANO_x$ emissions in the EU to NA regions (r =
0.6), and more weakly on the ratio of EU:NA AVOC emissions (r = 0.5; Supplemental Figure 2). The
correlation is even weaker between the ratio of PAN SRRs for these two regions with inter-model
differences in transport as diagnosed with the CO tracers from EU versus NA (r = 0.4).  At Mount
Waliguan, the strongest relationship is found for the ratio of AVOC emissions (r = 0.5; Supplemental
Figure 2).

We repeat this correlation analysis of inter-model differences in ratios of $ANO_x$ emissions, AVOC
emissions, or the idealized CO tracers of transport from a region, but for the ratio of PAN SRRs from two
intercontinental regions. At Mount Bachelor, the EU and EA source regions contribute similar amounts to
multi-model mean PAN during April (Figure 1). Across the individual models, however, the ratio of the EU
to EA source region on PAN at Mount Bachelor varies from less than half to a factor of two (Supplemental
Figure 3). We find that the ratio of PAN attributed to the EU versus EA source regions at Mount Bachelor
correlates strongly across the models with the ratio of the anthropogenic volatile organic compound
(AVOC) emissions in the respective source regions (r = 0.8; Supplemental Figure 3). In contrast, the ratio
of EU:EA anthropogenic emission influence on PAN at Mount Bachelor shows little correlation with the
respective regional $NO_x$ emissions used in the models, or with the differences in the simulated transport
tracers (r=0.3 for both cases). As at Mount Bachelor, the model spread in the contribution to total simulated
PAN from the EA versus NA source regions at both Jungfraujoch and Mount Waliguan depend most on the
regional AVOC ratios (r = 0.8 and 0.6, respectively; Supplemental Figure 3), with little correlation with
inter-model differences in NA:EA $ANO_x$ emissions. Some correlation also emerges between the NA:EA
source-receptor relationships for PAN and the NA:EA transport tracers (r = 0.6 at both sites; Supplemental
Figure 3). Finally, we do not find any obvious link between PAN SRRs and the choice of meteorological
fields (the individual symbols in Supplemental Figures 2 and 3).
**6. Linking PAN and $O_3$ SRRs**
We address here the extent to which observational constraints on PAN SRRs might also serve to narrow the
range of uncertainty in the inter-model spread in intercontinental SRRs for $O_3$ (e.g., Fiore et al., 2009). We
expect some commonality between the sensitivity of PAN and $O_3$ to changes in precursor emissions
because (1) both species are produced from chemical reactions involving $NO_x$ and VOC, and (2) PAN
serves as a $NO_x$ reservoir, which upon decomposition releases $NO_x$ that can then produce $O_3$ far downwind
of the region where the PAN (and $O_3$) precursors were originally emitted. Furthermore, earlier analysis of
HTAP1 ozone continental-scale SRRs also identified a correlation with the model AVOC emissions,
particularly over EU (Fiore et al., 2009).

We assess the extent to which the inter-model range in source region influence on mountaintop PAN levels
in April is relevant for interpreting $O_3$ SRRs by correlating PAN and $O_3$ SRRs at the three mountaintop
sites (Figure 7). Relationships vary across the individual source-receptor pairs, with the inter-model
variability in PAN explaining 16-60% of the inter-model differences in $O_3$ at the mountain sites. The
strongest relationships occur for the influence of regional sources at Mount Bachelor (from NA) and
Jungfraujoch (from EU). At Mount Waliguan, the EU and EA source-receptor relationships for PAN and
$O_3$ are of similar strength (r = 0.7). Intercontinental source-receptor pairs for $O_3$ and PAN at Mount
Bachelor and Mount Waliguan are also significant to within 90%, with variability in the PAN attributed to
intercontinental source regions explaining 25-35% and 30-45%, respectively, of the variability in the
corresponding $O_3$ SRRs.

We expand the correlation analysis of ozone and PAN SRRs from the free troposphere sampled at the
mountaintop sites to large-scale SRRs in surface air over the HTAP1 continental regions. Of the significant
relationships in Figure 7 (p < 0.10), 6 out of 7 also emerge as significant in Figure 8. We infer that
conclusions drawn from a limited number of mountaintop sites regarding PAN SRRs and their relationship
to ozone SRRs are relevant, at least according to the models, on much broader scales.

We repeat the analysis in Figure 5 but for $O_3$ to consider the influence of the three source regions on the
three mountaintop sites (nine total source-receptor pairs), but find little relationship between the model
spread in the simulated $O_3$ SRRs and in the magnitude of the regional AVOC or $ANO_x$ emissions. Model
differences in transport as diagnosed by the idealized regional CO tracers correlates more with $O_3$ SRRs
than for PAN for all source-receptor pairs, though the correlations remains weak except for COfromEU
with $O_3$ SRRs at Jungfraujoch. Overall, this analysis supports earlier findings that PAN is more sensitive to
changes in emissions (and subsequent chemistry), particularly for VOC precursors, than $O_3$.

The correlations between SRRs for PAN and $O_3$ could reflect a role for PAN transport in contributing to $O_3$
production over the receptor region, or may instead reflect co-production of PAN and $O_3$ from oxidation of
regional precursor emissions followed by transport in the same air mass. In the latter case, PAN is serving
as a proxy for $O_3$ transport whereas in the former case, PAN is serving as the actual pathway by which $O_3$ is
transported. We do not have model diagnostics that allow us to distinguish between these two roles for
PAN. The correlations between PAN and $O_3$ SRRs, however, suggest that long-term PAN measurements
contain signals relevant for constraining the relative importance of regional vs. intercontinental emissions
on both PAN and $O_3$. We examine the strength of these signals by correlating the $O_3$ SRRs at each site
with total PAN as simulated at each site. Relationships are far weaker than for the PAN SRRs and total
PAN shown in Figure 4, but correlations are significant between total PAN at Jungfraujoch for $O_3$ from EU
(r=0.67; p=0.03) and at Mount Bachelor for $O_3$ from NA (r=0.61; p=0.04; Supplemental Figure 4).
**7. Conclusions and recommendations**
Our proof of concept approach applies the HTAP1 multi-model ensemble to identify a strong inter-model
correlation between PAN source-receptor relationships (SRRs; defined as the difference in simulations with
20% emission reductions separately within each of the northern mid-latitude continents) and simulated total
PAN at mountaintop sites during April. Our findings imply promise for developing "emergent constraints"
(e.g., Hall and Qu, 2006; Borodina et al., 2017; Cox et al., 2018) from more routine PAN measurements to
narrow uncertainty in wide-ranging model estimates of PAN SRRs, quantities that are not directly
observable yet relevant to air quality policy (e.g., HTAP 2010). Inter-model correlations of the responses
of PAN versus $O_3$ to perturbations in regional anthropogenic emissions (Figures 7 and 8) imply that
constraints on PAN SRRs are relevant for lowering uncertainty in $O_3$ SRR estimates. This connection
between PAN and $O_3$ likely reflects the dual role of PAN as both a pathway for $O_3$ transport (by producing
$O_3$ upon its decomposition following transport), and as a proxy for $O_3$ transport (as it is produced alongside
$O_3$ in the polluted continental boundary layer).

Establishing the strongest constraints possible on simulated SRRs for PAN and $O_3$ will require (1)
measurements and simulations with chemical transport models that coincide, and (2) a sufficiently long
measurement record to build a climatology suitable for evaluating chemistry-climate models that generate
their own meteorology. Repeated sampling for the month of April may be sufficient to provide constraints
on model responses to changes in anthropogenic emissions. PAN measurements over multiple seasons are
necessary to evaluate model responses of PAN to climate change (e.g., by changing temperature and
weather-sensitive precursor emissions) and the resulting influence on atmospheric $O_3$ and oxidizing
capacity (e.g., Doherty et al., 2013). For example, changes in meteorology and biomass burning (Fischer et
al., 2011; Zhu et al., 2015) such as those driven by ENSO (Koumoutsaris et al., 2008), as well as biogenic
and lightning sources (Payne et al., 2017) vary from year to year and are expected to change as climate
warms.

We identified only five multi-year datasets at mountain sites, four of which are located near each other in
Europe, and only one of which continues at present (Schauinsland). Our analysis suggests that future
measurements at Mount Waliguan would provide constraints on PAN SRRs, particularly for PAN
originating in EA (Figure 4). Additional work could systematically examine over 60 stations at altitudes
above 2500 m in the Tropospheric Ozone Assessment Report (TOAR) database (Schultz et al., 2017).
We recommend archival of daily model fields for future applications of this multi-model emergent
constraint approach to SRRs. Access to daily model fields permits (1) a more rigorous process-oriented
evaluation of specific events (e.g., Fischer et al., 2010; Alvarado et al., 2010; Arnold et al., 2015), and (2)
comparison with satellite-derived tropospheric PAN columns, which show promise for documenting PAN
distributions, particularly in the upper troposphere, and their temporal variability and spatial patterns across
the globe (e.g., Fadnavis et al., 2014; Jiang et al., 2016; Payne et al., 2014; 2017; Zhu et al., 2015; 2017).
We also suggest archiving daily tracers tagged by emission region to isolate the role of model differences in
transport during individual events. In addition, Lin et al. (2017) have demonstrated that applying a filtering
technique based on daily idealized CO regional tracers can better isolate free tropospheric air from surface
air masses when comparing coarse resolution models with high altitude measurements.
By focusing on April, our analysis largely minimizes complexities introduced by inter-model differences in
biogenic, fire, and lightning sources that further complicate disentangling summertime discrepancies in
simulated PAN and $O_3$ (e.g., Arnold et al., 2015; Emmons et al., 2015) and restricts inter-model differences
to those associated with anthropogenic emissions and the subsequent chemistry and transport.
Nevertheless, we find a wide range in inter-model SRR relationships that reflects uncertainties in emissions
and different model representations of VOC chemistry, including PAN yields from VOCs (Figure 5;
Emmerson and Evans, 2009; Fischer et al., 2014; Arnold et al., 2015; Emmons et al., 2015; Knote et al.,
2015). Future multi-model efforts could seek to parse separately the influence of differences in total
anthropogenic VOC emissions, the mix of emitted VOC species and their reactivity, and the chemical
production of PAN and $O_3$. Documenting these aspects of model configuration would help to establish
benchmarks for inter-model differences in simulated total PAN, $O_3$, and their SRRs, against which future
model simulations (and multi-model ensembles) can be assessed.
**Acknowledgments**
We thank Mathew Evans (York University, UK) and Terry Keating (U.S. EPA) for useful discussions, and
two anonymous referees for their constructive comments. AMF and BND acknowledge NASA MAP
(NNX14AM38G). DSS acknowledges NERC (grants NE/K001329/1 and NE/N003411/1) and the
ARCHER UK National Supercomputing Service (http://www.archer.ac.uk). Data for the Mt. Bachelor
Observatory are archived and available at the University of Washington data archive:
https://digital.lib.washington.edu/researchworks/browse?type=subject&value=Mt.+Bachelor+Observatory.
The PAN data for the European Mountain sites is archived by the World Data Centre for Greenhouse Gases
(http://ds.data.jma.go.jp/gmd/wdcgg/). Upon publication, the data used to generate the figures will be
placed in a CSU digital repository that we have already established for this manuscript
(https://hdl.handle.net/10217/185610). This is Lamont contribution number 8251.

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

Table 1 published in Vol. 106 (D17), p.20507).

Koumoutsaris, S., Bey, I., Generoso, S., and Thouret, V.: Influence of El Niño-Southern Oscillation on the
interannual variability of tropospheric ozone in the northern midlatitudes, J. Geophys. Res., 113,
D19301, 10.1029/2007jd009753, 2008.

Kuhn, M., Builtjes, P. J. H., Poppe, D., Simpson, D., Stockwell, W. R., Andersson-Sköld, Y., Baart, A., Das, M., Fiedler, F., Hov, Ø., Kirchner, F., Makar, P. A., Milford, J. B., Roemer, M. G. M., Ruhnke, R., Strand, A., Vogel, B., and Vogel, H.: Intercomparison of the gas-phase chemistry in several chemistry and transport models, Atmospheric Environment, 32, 693-709, 1998.

Liang, J., Horowitz, L. W., Jacob, D. J., Wang, Y., Fiore, A. M., Logan, J. A., Gardner, G. M., and Munger, J. W.: Seasonal variations of reactive nitrogen species and ozone over the United States and export fluxes to the global atmosphere, Journal of Geophysical Research, 103, 13,435-413,450, 1998.

Lin, M., Horowitz, L. W., Payton, R., Fiore, A. M., and Tonnesen, G.: US surface ozone trends and extremes from 1980 to 2014: quantifying the roles of rising Asian emissions, domestic controls, wildfires, and climate, Atmos. Chem. Phys., 17, 2943-2970, https://doi.org/10.5194/acp-17-2943-2017, 2017.

Lin, M., Holloway, T., Carmichael, G. R., and Fiore, A. M.: Quantifying pollution inflow and outflow over East Asia in spring with regional and global models, Atmos. Chem. Phys., 10, 4221-4239, 10.5194/acp-10-4221-2010, 2010.

Liu, S. C., Trainer, M., Fehsenfeld, F. C., Parrish, D. D., Williams, E. J., Fahey, D. W., Hubler, G., and Murphy, P. C.: Ozone production in the Rural Troposphere and the Implications for Regional and Global Ozone Distributions, Journal of Geophysical Research, 92, 4191-4207, 1987.

Moxim, W. J., Levy, H., II, and Kasibhatla, P. S.: Simulated global tropospheric PAN: Its transport and impact on NOx, J. Geophys. Res., 101, 12621-12638, 10.1029/96jd00338, 1996.

Orbe, C., D. W. Waugh, H. Yang, J.-F. Lamarque, S. Tilmes, and D. E. Kinnison (2017), Tropospheric transport differences between models using the same large-scale meteorological fields, Geophys. Res. Lett., 44, 1068–1078, doi:10.1002/2016GL071339.

Pandey Deolal, S., Henne, S., Ries, L., Gilge, S., Weers, U., Steinbacher, M., Staehelin, J., and Peter, T.: Analysis of elevated springtime levels of Peroxyacetyl nitrate (PAN) at the high Alpine research sites Jungfraujoch and Zugspitze, Atmos. Chem. Phys., 14, 12553-12571, 10.5194/acp-14-12553-2014, 2014.

Pandey Deolal, S., Staehelin, J., Brunner, D., Cui, J., Steinbacher, M., Zellweger, C., Henne, S., and Vollmer, M. K.: Transport of PAN and NOy from different source regions to the Swiss high alpine site Jungfraujoch, Atmospheric Environment, 64, 103-115, https://doi.org/10.1016/j.atmosenv.2012.08.021, 2013.

Payne, V. H., Alvarado, M. J., Cady-Pereira, K. E., Worden, J. R., Kulawik, S. S., and Fischer, E. V.: Satellite observations of peroxyacetyl nitrate from the Aura Tropospheric Emission Spectrometer, Atmos. Meas. Tech., 7, 3737-3749, 10.5194/amt-7-3737-2014, 2014.

Payne, V. H., Fischer, E. V., Worden, J. R., Jiang, Z., Zhu, L., Kurosu, T. P., and Kulawik, S. S.: Spatial variability in tropospheric peroxyacetyl nitrate in the tropics from infrared satellite observations in 2005 and 2006, Atmos. Chem. Phys., 17, 6341-6351, 10.5194/acp-17-6341-2017, 2017.

Penkett, S. A., and Brice, K. A.: The spring maximum in photo-oxidants in the Northern Hemisphere
troposphere, Nature, 319, 655-657, 1986.

Roberts, J. M.: PAN and Related Compounds, in: Volatile Organic Compounds in the Atmosphere, edited
by: Koppmann, R., Blackwell Publishing, 500, 2007.

Schmitt, R., and Volz-Thomas, A.: Climatology of Ozone, PAN, CO, and NMHC in the Free Troposphere
Over the Southern North Atlantic, Journal of Atmospheric Chemistry, 28, 245-262,
10.1023/A:1005801515531, 1997.

Schultz, M. G. et al.: On the origin of tropospheric ozone and NOx over the tropical South Pacific, J.
Geophys. Res., 104, D5, 5829-5843, 1999.

Schultz, M., Rast, S., van het Bolscher, M., Pulles, T., Brand, R., Pereira, J., Mota, B., Spessa, A.,
Dalsoren, S., van Noije, T., and Szopa, S.: Emission data sets and methodologies for estimating
emissions, Hamburg, 2007.

Schultz, M. G., Heil, A., Hoelzemann, J. J., Spessa, A., Thonicke, K., Goldammer, J. G., Held, A. C.,
Pereira, J. M. C., and van het Bolscher, M.: Global wildland fire emissions from 1960 to 2000, Global
Biogeochemical Cycles, 22, GB2002, 10.1029/2007GB003031, 2008.

Schultz MG, Schröder S, Lyapina O, Cooper O, Galbally I, Petropavlovskikh I, et al.: Tropospheric Ozone
Assessment Report: Database and Metrics Data of Global Surface Ozone Observations, Elem Sci
Anth., 5, 58, DOI: http://doi.org/10.1525/elementa.244, 2017.

Shindell, D. T., Chin, M., Dentener, F., Doherty, R. M., Faluvegi, G., Fiore, A. M., Hess, P., Koch, D. M.,
MacKenzie, I. A., Sanderson, M. G., Schultz, M. G., Schulz, M., Stevenson, D. S., Teich, H., Textor,
C., Wild, O., Bergmann, D. J., Bey, I., Bian, H., Cuvelier, C., Duncan, B. N., Folberth, G., Horowitz,
684        L. W., Jonson, J., Kaminski, J. W., Marmer, E., Park, R., Pringle, K. J., Schroeder, S., Szopa, S.,
Takemura, T., Zeng, G., Keating, T. J., and Zuber, A.: A multi-model assessment of pollution transport
to the Arctic, Atmos. Chem. Phys., 8, 5353-5372, 10.5194/acp-8-5353-2008, 2008.

Singh, H. B.: Reactive nitrogen in the troposphere, Environmental Science and Technology, 21, 320-327,
1987.

Singh, H. B., and Hanst, P. L.: Peroxyacetyl nitrate (PAN) in the unpolluted atmosphere: An important
reservoir for nitrogen oxides, Geophysical Research Letters, 8, 941-944, 1981.

Singh, H. B., and Salas, L. J.: Measurements of peroxyacetyl nitrate (pan) and peroxypropionyl nitrate
(ppn) at selected urban, rural and remote sites, Atmospheric Environment (1967), 23, 231-238,
https://doi.org/10.1016/0004-6981(89)90115-7, 1989.

Stevenson, D.S., et al.: Multimodel ensemble simulations of present-day and near-future tropospheric
ozone, J. Geophys. Res., 111, D08301, doi:10.1029/2005JD006338, 2006.

Turnipseed, A. A., Huey, L. G., Nemitz, E., Stickel, R., Higgs, J., Tanner, D. J., Slusher, D. L., Sparks, J.
P., Flocke, F., and Guenther, A.: Eddy covariance fluxes of peroxyacetyl nitrates (PANs) and NOy to a
coniferous forest, Journal of Geophysical Research: Atmospheres, 111, D09304,
10.1029/2005jd006631, 2006.

Val Martin, M., Honrath, R. E., Owen, R. C., and Lapina, K.: Large-scale impacts of anthropogenic
pollution and boreal wildfires on the nitrogen oxide levels over the central North Atlantic region,
Journal of Geophysical Research, 113, doi:10.1029/2007JD009689, 2008.
van der Werf, G. R., Randerson, J. T., Giglio, L., Collatz, G. J., Kasibhatla, P. S., and Arellano Jr, A. F.:
Interannual variability in global biomass burning emissions from 1997 to 2004, Atmos. Chem. Phys.,
6, 3423-3441, 10.5194/acp-6-3423-2006, 2006.
van der Werf, G. R., Randerson, J. T., Giglio, L., Collatz, G. J., Mu, M., Kasibhatla, P. S., Morton, D. C.,
DeFries, R. S., Jin, Y., and van Leeuwen, T. T.: Global fire emissions and the contribution of
deforestation, savanna, forest, agricultural, and peat fires (1997–2009), Atmos. Chem. Phys., 10,
11707-11735, 10.5194/acp-10-11707-2010, 2010.
Volz-Thomas, A., Xueref, I., and Schmitt, R.: An automatic gas chromatograph and calibration system for
ambient measurements of PAN and PPN, Environmental Science and Pollution Resources, Special
Issue 4, 72-76, 2002.
Wang, Y., and Jacob, D. J.: Anthropogenic forcing on tropospheric ozone and OH since preindustrial
times, Journal of Geophysical Research, 103, 31,123-131,135, 1998.
Warneck, P., and Zerbach, T.: Synthesis of peroxyacetyl nitrate by acetone photolysis, Environmental
Science and Technology, 26, 74-79, 1992.
Weiss-Penzias, P., Jaffe, D. A., Jaeglé, L., and Liang, Q.: Influence of long-range-transported pollution on
the annual and diurnal cycles of carbon monoxide and ozone at Cheeka Peak Observatory, Journal of
Geophysical Research: Atmospheres, 109, n/a-n/a, 10.1029/2004JD004505, 2004.
Wild, O., Law, K., McKenna, D., Bandy, B., Penkett, S., Pyle, J.: Photochemical trajectory modeling
studies of the North Atlantic region during August 1993, Journal of Geophysical Research:
Atmospheres. 101, D22, p. 29269-29288, 1996.
Wild, O., Fiore, A. M., Shindell, D. T., Doherty, R. M., Collins, W. J., Dentener, F. J., Schultz, M. G.,
Gong, S., MacKenzie, I. A., Zeng, G., Hess, P., Duncan, B. N., Bergmann, D. J., Szopa, S., Jonson, J.
E., Keating, T. J., and Zuber, A.: Modelling future changes in surface ozone: a parameterized
approach, Atmos. Chem. Phys., 12, 2037-2054, 10.5194/acp-12-2037-2012, 2012.
Wu, S., Duncan, B. N., Jacob, D. J., Fiore, A. M., and Wild, O.: Chemical nonlinearities in relating
intercontinental ozone pollution to anthropogenic emissions, Geophys. Res. Lett., 36, L05806,
10.1029/2008gl036607, 2009.
Xue, L. K., Wang, T., Zhang, J. M., Zhang, X. C., Deliger, Poon, C. N., Ding, A. J., Zhou, X. H., Wu, W.
S., Tang, J., Zhang, Q. Z., and Wang, W. X.: Source of surface ozone and reactive nitrogen speciation
at Mount Waliguan in western China: New insights from the 2006 summer study, J. Geophys. Res.,
116, D07306, doi:10.1029/2010JD014735, 2011.
Zellweger, C., Ammann, M., Buchmann, B., Hofer, P., Lugauer, M., Rüttimann, R., Streit, N.,
Weingartner, E., and Baltensperger, U.: Summertime NOy speciation at the Jungfraujoch, 3580 m asl,
Switzerland, Journal of Geophysical Reserach, 105, 2000.
Zellweger, C., Forrer, J., Hofer, P., Nyeki, S., Schwarzenbach, B., Weingartner, E., Ammann, M., and
Baltensperger, U.: Partitioning of reactive nitrogen ($NO_y$) and dependence on meteorological
conditions in the lower free troposphere, Atmospheric Chemistry and Physics, 3, 779-796, 2003.
Zhang, L., Jacob, D. J., Boersma, K. F., Jaffe, D. A., Olson, J. R., Bowman, K. W., Worden, J. R.,
Thompson, A. M., Avery, M. A., Cohen, R. C., Dibb, J. E., Flocke, F. M., Fuelberg, H. E., Huey, L.
G., McMillian, W. W., Singh, H. B., and Weinheimer, A. J.: Transpacific transport of ozone pollution
and the effect of recent Asian emission increases on air quality in North America: an integrated
analysis using satellite, aircraft, ozonesonde, and surface observations, Atmospheric Chemistry and
Physics, 8, 6117-6136, 2008.
Zhu, L., V. H. Payne, T. W. Walker, J. R. Worden, Z. Jiang , S. S. Kulawik, and E. V. Fischer (2017),
PAN in the eastern Pacific free troposphere: A satellite view of the sources, seasonality, interannual
variability, and timeline for trend detection, J. Geophys. Res. Atmos., 122, 3614–3629,
doi:10.1002/2016JD025868.
Zhu, L., Fischer, E. V., Payne, V. H., Worden, J. R., and Jiang, Z.: TES observations of the interannual
variability of PAN over Northern Eurasia and the relationship to springtime fires, Geophysical
Research Letters, 42, 7230-7237, 10.1002/2015GL065328, 2015.

**Table 1: Models contributing to the HTAP1 simulations (SR1, SR6xx, and COfromXX) used in this study.**

| Model | Resolution (lat-lon-layers) | Institute | Model contact | SR1 | SR6xx | COfrom xx | Plotting symbol |
|---|---|---|---|---|---|---|---|
| CAMCHEM-3311m13 | 2.5°x2°x 30 | NCAR, USA | Peter Hess | X | X | X | Filled circle |
| FRSGCUCI-v01 | 2.81°x2.81°x37 | Lancaster Univ., UK | Oliver Wild | X | X | X | Filled upward triangle |
| GEMAQ-v1p0 | 2°x2°x28 | York Univ., Canada | Alex Lupu | X | X | X | Filled downward triangle |
| GEOSChem-v07 | 2.5°x2°x 30 | Harvard Univ., USA | Rokjin Park | X | X | X | Filled diamond |
| GISS-PUCCINI-modelE | 5°x4°x23 | NASA GISS, USA | Drew Shindell | X | X | X | Filled square |
| GMI-v02f | 2.5°x2°x 42 | NASA GSFC, USA | Bryan Duncan | X | X | X | Open circle |
| LMDZ3-INCA1 | 3.75° x 2° x 19 | CEA, France | Sophie Szopa | X | X | | Open upward triangle |
| LLNL-IMPACT-T5a | 2.5° x 2° x 48 | LLNL, USA | Dan Bergmann | X | | | Open downward triangle |
| MOZARTGFDL-v2 | 1.88° x 1.88° x 28 | NOAA GFDL, USA | Arlene Fiore | X | X | X | Open diamond |
| MOZECH-v16 | 1.88° x 1.88° x 28 | FZ Julich, Germany | Martin Schultz | X | X | X | Open square |
| STOC-HadAM3-v01 | 5° x 5° x 19 | University of Edinburg, UK | Ruth Doherty, David Stevenson | X | X | X | Plus sign |
| STOCHEM-v02 | 3.75 x 2.5° x 20 | Met Office, Hadley Center, UK | Bill Collins, Michael | X | | | X |

| | | | | | | | |
|---|---|---|---|---|---|---|---|
| | | | Sanderson | | | | |
| TM5-JRC-cy2-ipcc-v1 | 1º x 1º x 25 | JRC, Italy | Frank Dentener | X | X | X | Filled right facing triangle |
| UM-CAM-v01 | 3.75º x 2.5º x 19 | University of Cambridge, UK | Guang Zeng | X | X | X | Filled left facing triangle |














**Table 2: Simulations from HTAP1 used in this study.**

| Simulation | Description |
| --- | --- |
| SR1 | Base case (see Section 2.1 for details) |
| SR6EA | SR1 but with anthropogenic emissions of all $O_3$ precursors ($NO_x$+CO+NMVOC) and aerosols within EA decreased by 20% |
| SR6EU | SR1 but with 20% emissions reductions within the EU region |
| SR6NA | SR1 but with 20% emissions reductions within the NA region. |
| COfromEA | Idealized tracer simulation in which all models use identical CO emissions, emitted within the EA region, with a 50-day e-folding lifetime. |
| COfromEU | Same as COfromEA but for the EU region. |
| COfromNA | Same as COfromEA but for the NA region. |






**Table 3: Mountaintop sites with multiple years of PAN observations used in this study.**

| Site | Location | Elevation | Measurement Period (s) | Reference (s) |
|------|----------|-----------|------------------------|---------------|
| Mount Bachelor | 43.979° N, 121.687° W | 2763m | 3 April – 18 June 2008, 30 August – 7 October 2008, 26 March – 20 May 2009, 23 March – 25 May 2010 | (Fischer et al., 2010;Fischer et al., 2011) |
| Hohenpeissenberg | 47.80° N, 11.02° E | 985 m | January 2003 – December 2008 | http://www.dwd.de/de/GAW (Gilge et al., 2010) |
| Jungfraujoch | 46.55°N, 7.98°E | 3580 m | April 1997 – May 1998, Aug 30 2005 – Sept 16 2005, Throughout 2005, but not continuous | (Balzani Lööv et al., 2008;Carpenter et al., 2000;Zellweger et al., 2000;Zellweger et al., 2003) |
| Zugspitze | 47.42° N, 10.98° E | 2960 m | May 2004 – December 2008 | http://gaw.kishou.go.jp |
| Schauinsland | 47.92° N, 7.92° E | 1205m | January 1995 – December 2010 | www.umweltbundesamt.de |



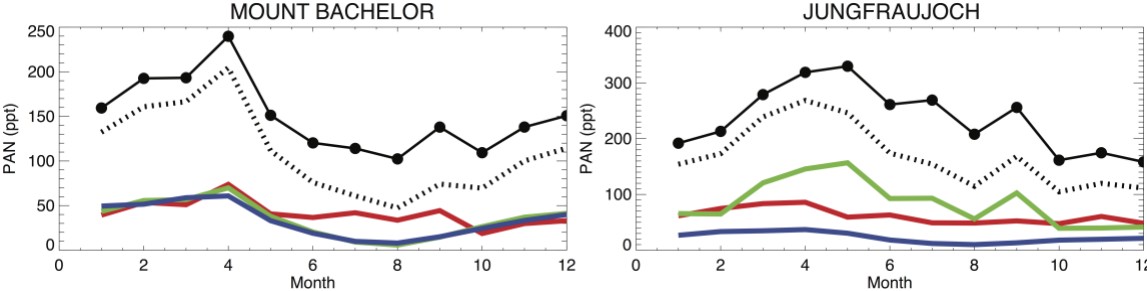


**Figure 1: Multi-model monthly mean total PAN mixing ratios (black circles and solid lines) at Mount Bachelor (left) and Jungfraujoch (right). We take the difference between the base simulation (SR1) and one in which emissions are decreased by 20% and then multiply the difference by 5 to estimate a 100% contribution associated with anthropogenic precursor emissions from Europe (green), North America (red), East Asia (blue). The sum of the anthropogenic contribution from these three regions is shown (dashed black) for comparison with total simulated PAN.**

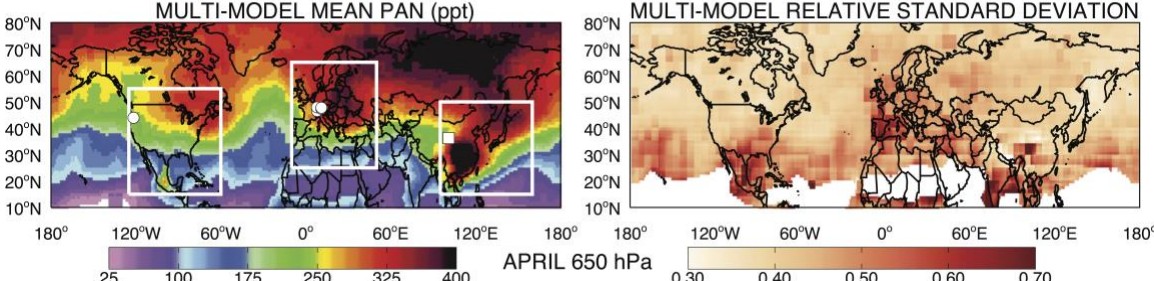

786

**Figure 2: Multi-model ensemble (n=14; Table 1) average PAN mixing ratios (ppt; left panels) and relative standard deviation (the absolute standard deviation across the models divided by the ensemble mean; right panels) at 650 hPa in April; relative standard deviations are masked out (white) for regions where multi-model mean PAN falls below 100 ppt. The models were sampled at 650 hPa by vertically interpolating between the bounding grid cells and then re-gridded horizontally to a common 1°x1° grid. White lines denote the HTAP1 source regions: North America (NA), Europe and North Africa (EU), and East Asia (EA) from left to right. White circles indicate the five mountain sites with multi-year PAN observations used in our analysis (note: Zugspitze and Hohenpeissenberg are too close to differentiate on the map; see Table 3). Mount Waliguan in Asia, where we lack multi-year measurements but conduct model analysis, is denoted by the white square.**

796

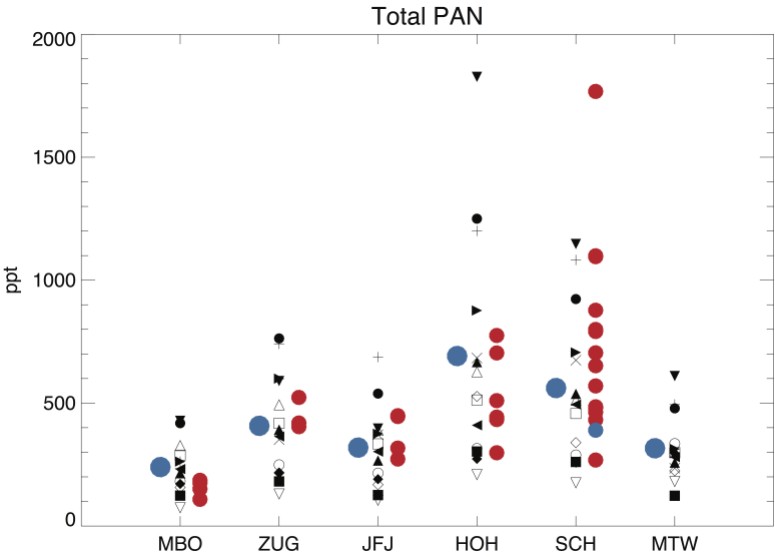

797

**Figure 3. April mean PAN abundances (ppt) simulated (black symbols, one per model as defined in Table 1; blue circles offset to the left show multi-model mean values) and measured (red circles offset to the right of the model values) at northern mid-latitude mountaintop sites: Mount Bachelor (MBO), Zugspitze (ZUG), Jungfraujoch (JFJ), Hohenpeissenberg (HOH), Schauinsland (SCH) and Mount Waliguan (MTW). The observed year 2001 April mean, which corresponds to the meteorological year used by most of the models, at Schauinsland is shown in blue to the right of the models.**

804

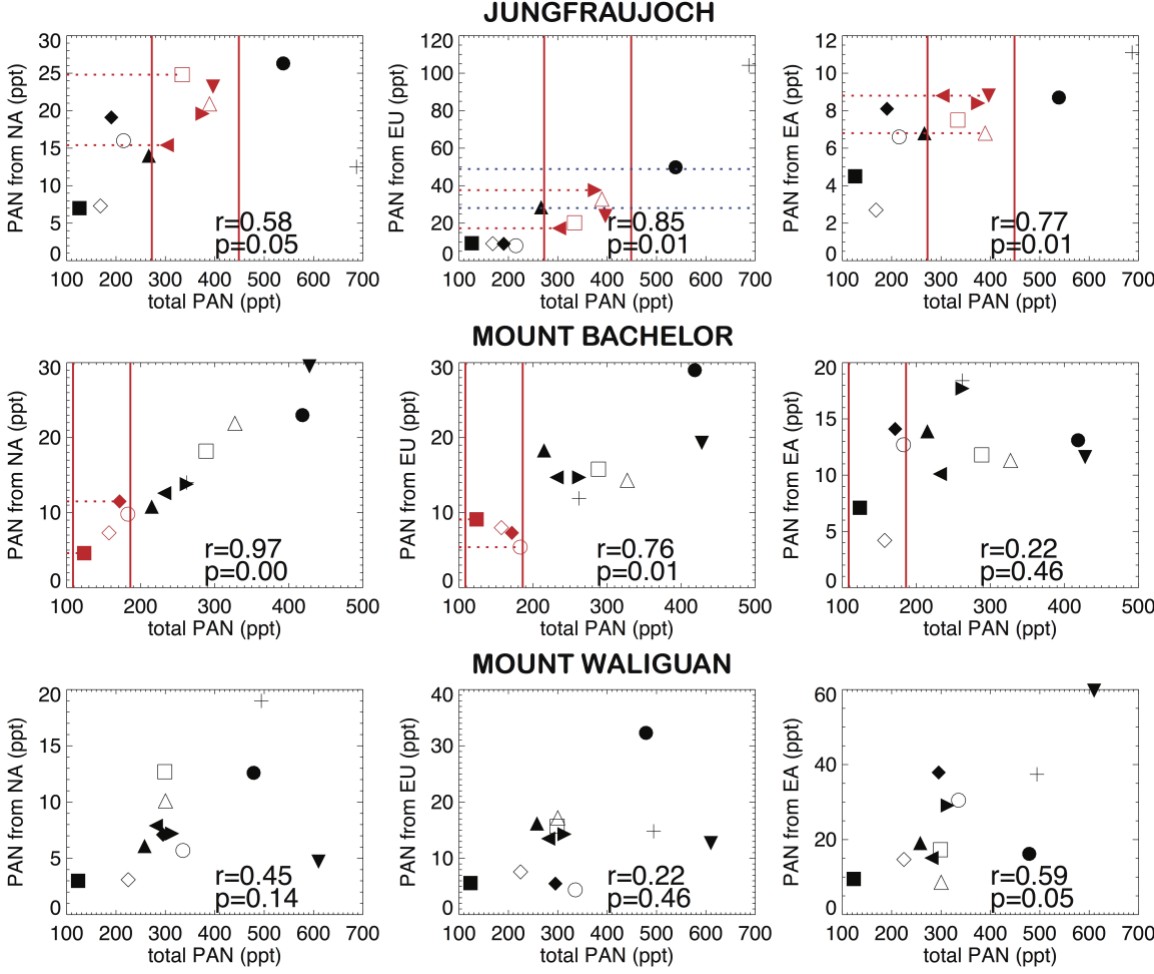

805

**Figure 4.** Simulated total PAN versus source-receptor relationships (SRRs) at each of three northern mid-
latitude sites. For Jungfraujoch and Mount Bachelor, vertical red lines bound the observed range in total PAN.
For source-receptor pairs with significant correlations (p ≤ 0.05), models falling within the observed range
(across years) are colored red, and horizontal red dashed lines extend to the ordinate, representing the emergent
constraint (narrower range resulting from selecting only those models falling in the observed range of total
PAN). At Jungfraujoch, the range (across years) in PAN attributed to the EU source region by back-trajectory
analysis (Pandey Deolal et al., 2013) is indicated by horizontal dashed blue lines. Individual models are denoted
by the symbols defined in Table 1.

814

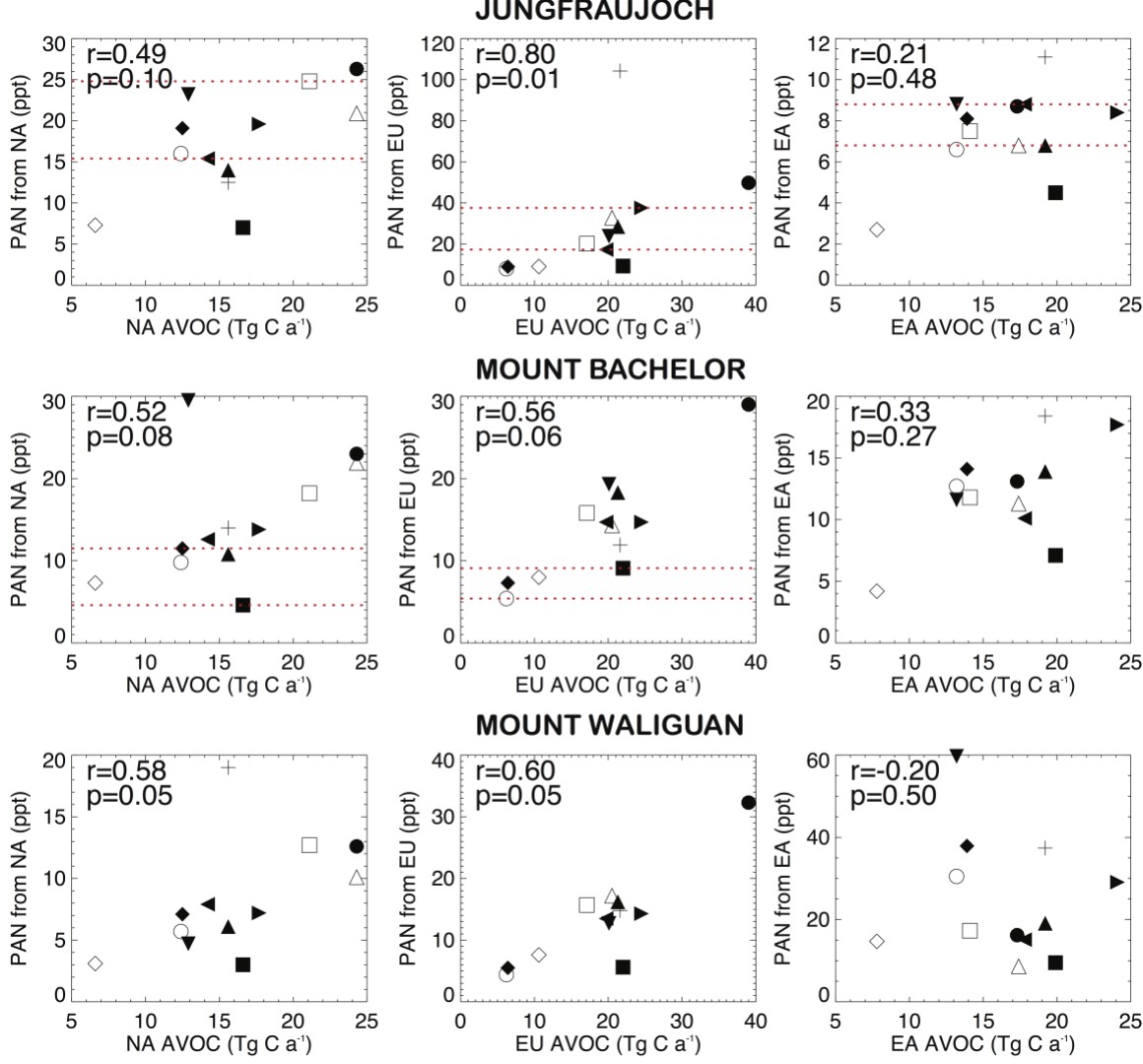

815

**Figure 5: SRRs diagnosed as the difference between the SR1 and SR6xx simulations in Table 1 for PAN (ppt) at Jungfraujoch (top), Mount Bachelor (middle), and Mount Waliguan (bottom) in each HTAP1 model (see Table 1 for symbol assigned to each model) versus the annual emission of anthropogenic VOC (AVOC; Tg C a$^{-1}$) within the NA (left), EU (middle) and EA (right) source regions. The Spearman rank correlation coefficient (more robust to outliers than the traditional Pearson coefficient) and associated p-value are shown in each panel. The horizontal red lines correspond to the values identified with the red symbols in Figure 4.**

822

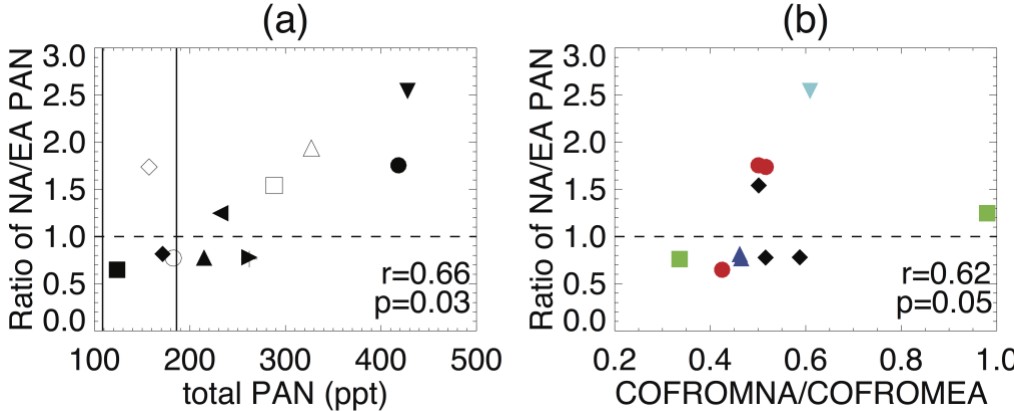

823

**Figure 6: Ratio of the PAN response to 20% emission reductions within NA versus EA plotted against (a) total**
**PAN and (b) the ratio of idealized tracers of model transport emitted from NA versus EA**
**(COfromNA/COfromEA; see Table 2) at Mount Bachelor as simulated by the HTAP1 models. Each symbol in**
**(a) represents a model as defined in Table 1; the range of observed total PAN at Mount Bachelor is indicated by**
**the black vertical lines. The colored symbols in (b) represent the meteorological fields used in the simulation:**
**blue triangles for GEOS winds; red circles for NCEP; black diamonds for ECMWF; cyan upside-down triangles**
**for CMC; green squares for general circulation models forced by observed sea surface temperatures and sea ice.**
**Both panels show Spearman rank correlation coefficients and p-values, as well as a black dashed horizontal line**
**at 1 to separate the models suggesting a higher NA influence (above) versus higher EA influence (below) on PAN**
**SRRs.**

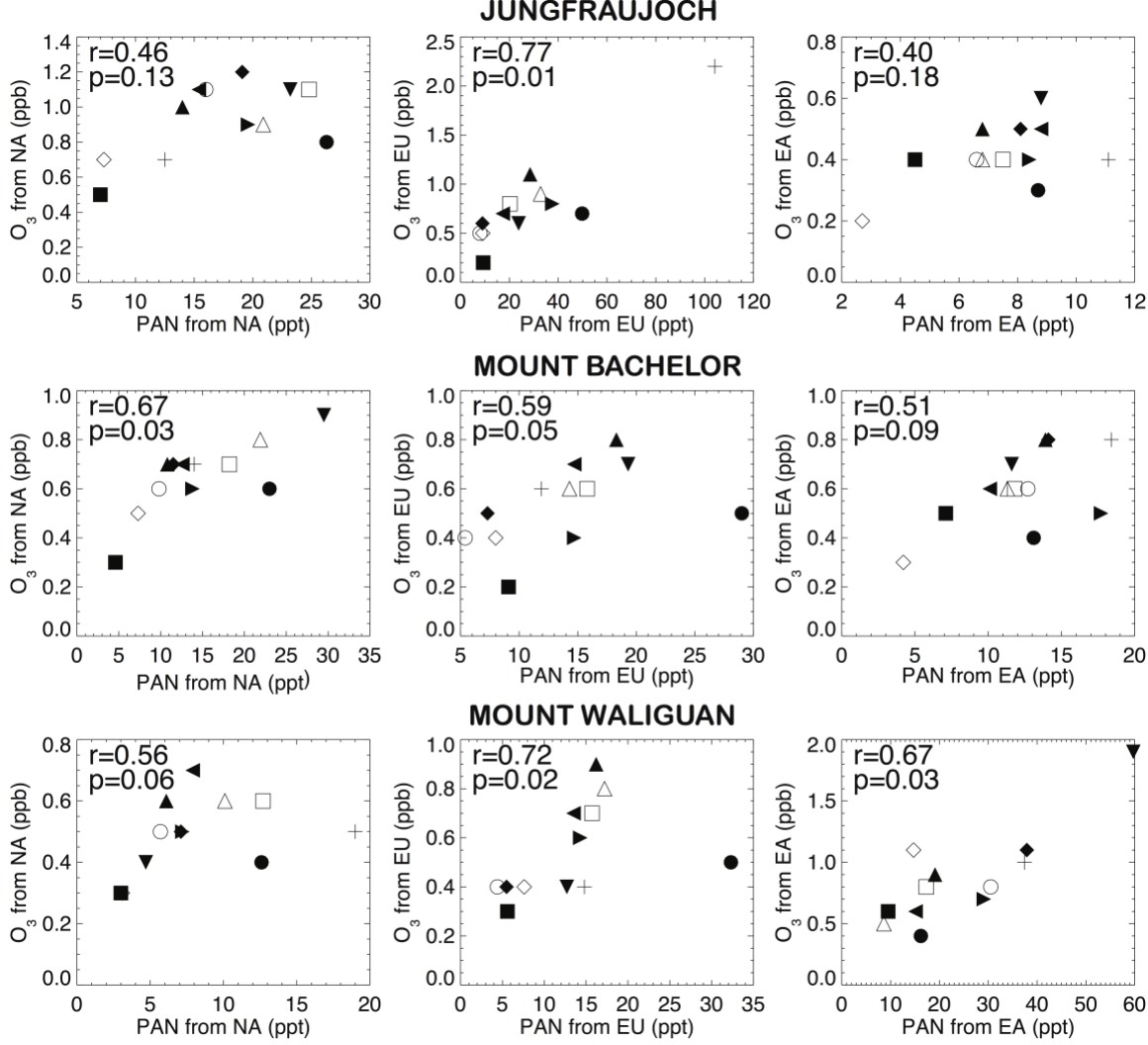


**Figure 7: SRRs for O₃ versus PAN at Jungfraujoch (top), Mount Bachelor (middle), and Mount Waliguan**
**(bottom), obtained by subtracting the SR6XX from the SR1 simulations (Table 2) available from 12 models,**
**where XX denotes the NA (left), EU (middle) or EA (right) source region. Each model thus contributes one point**
**(symbols defined in Table 1) in each panel. Spearman (rank) correlation coefficient and p-values are also shown.**

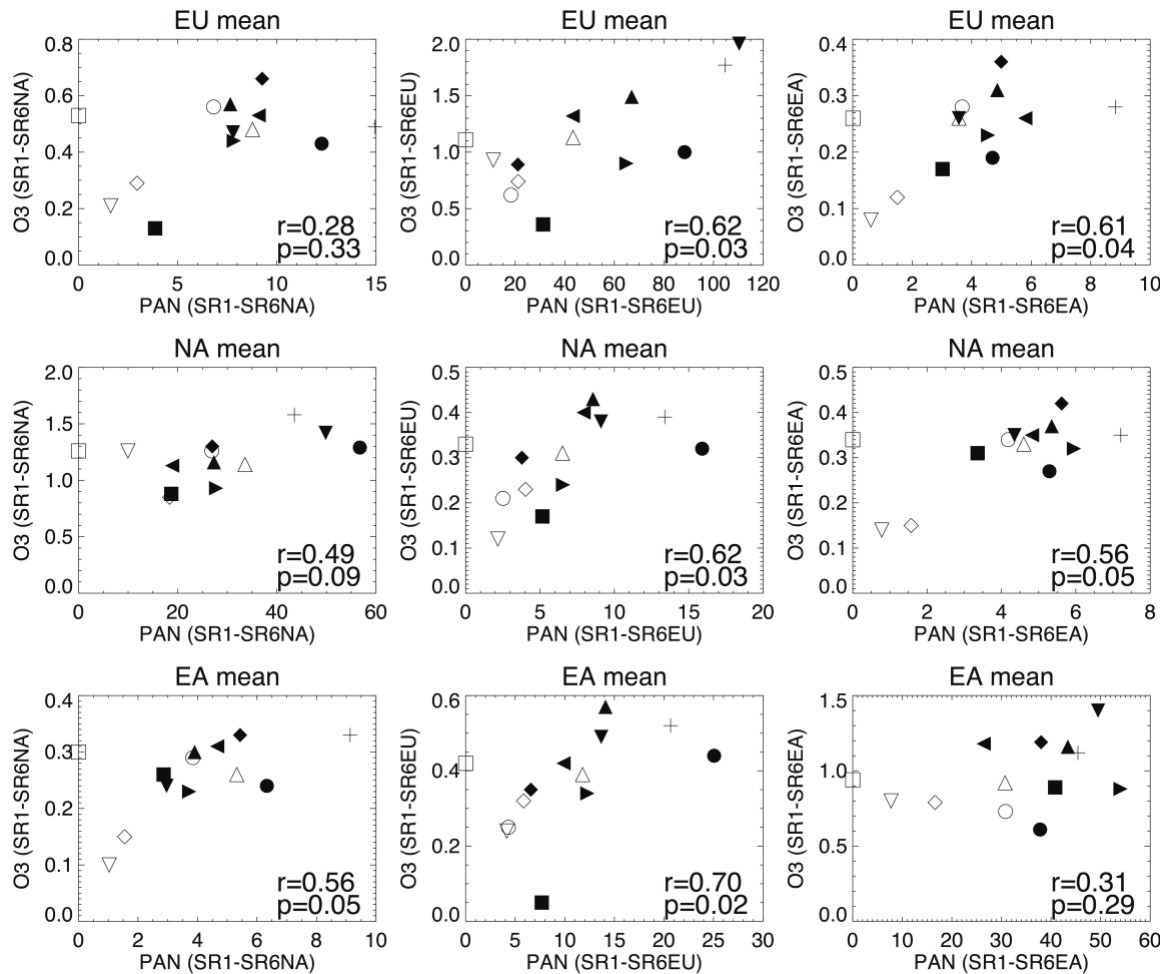


**Figure 8: SRRs for O₃ versus PAN in surface air over each of the HTAP1 northern mid-latitude continental regions: EU (top), NA (middle), and EA (bottom), obtained by subtracting the SR6XX from the SR1 simulations, where XX denotes the NA (left), EU (middle) or EA (right) source region. Spearman (rank) correlation coefficient and p-values are also shown. Symbols denote individual models as defined in Table 1. STOC-HadAM3-v01 is excluded here as an outlier that artificially raised the correlation significance.**