# Peer review of "Peroxy acetyl nitrate (PAN) measurements at northern 1"

_Atmospheric Chemistry and Physics, 2018_

## Referee Comment (RC1) · Anonymous Referee #1 · 18 Apr 2018

This paper attempts to understand the seasonal scale processes controlling the concentration of PAN at 5 mountain top sites in the northern mid-latitudes and to then use this as a tool to understand the processes controlling tropospheric O3. It attempts to do this through the framework of the HTAP1 model inter-comparison exercise, a Lagrangian modelling approach and the measurements made at the sites.

The paper is relatively long and often feels a bit winding. It has a large number of authors and this committee approach is obvious in the paper. For all the effort that has gone into the paper my major concern is that it's not obvious what has been learnt and whether it is new and / or interesting? From the abstract the major conclusions

seem to be: there are some differences between the attribution of the Eulerian and the Lagrangian approaches for a single site in Switzerland; VOCs are important for PAN production; ozone and PAN chemistries are qualitatively linked in models. I don't think any of these are particularly novel. I think however, that with some extra work this paper could provide some interesting insights.

These model simulations are old. They were run over a decade ago. How useful is this exercise if the models have now changed substantially? Has anything been learnt over the last decade which should be considered here. I don't think the authors can just ignore this issue.

One concern is that the paper investigates the importance of a number of issues in determining the processing controlling PAN for these mountaintop sites, but it makes little effort to then translate this understanding onto a larger spatial scale. How does PAN from EU VOCs make its way round the world? How strong is the PAN O3 relationship in different models globally? The paper discusses essentially the two sites that it has (one in the US and the multiple ones in almost exactly the same space in the Alps) and then seems to stop abruptly without widening its thinking to a more global or even hemispheric scale. What has been learnt which is more globally or regionally explicable? It would seem like a sensible next step to investigate the wider implications of the perturbation studies.

I'm not sure of the usefulness of the Lagrangian approach in this paper. PAN's lifetime varies significantly with temperature, and whether it being produced or lost depending upon a complex set of chemical reactions. The Langrangian approach may have some usefulness when looking at relatively short-lived, chemically simple tracers but it is not obvious that it has value when looking at PAN. The methodology is not described in any details and issues to do with subgrid processes (convection and boundary layer mixing) are not discussed at all. The authors need to show better its usefulness to PAN and then if they continue to use the measurements they should do something more useful with these calculatins. At the moment they show that the Lagrangian

model gives different results than the Eulerian models and then appear to provide a very handwaving route to get to some form of 'consistency' between the different modelling framworks. At this point they essentially then proceed ignoring the issues raised by the Lagrangian method. The paper would come to the same conclusions without Langrangian section. I would therefore suggest it was removed.

The authors argue that there needs to be more measurements made from mountain top sites. I'm not convinced. Especially in the Alps there is a raft of data from these sites. It may be that making more mountain top measurements from other locations on other continents would be useful, but the authors provide little evidence for that. Where would they think that these measurements should be made? Himalayas, Urals, White Mountains, Rockies? Presumably from their model simulations they could make some suggests.

Where there are interesting results (notably Figures 6 and 9) the authors don't really delve into the details. What explains the difference in AVOC emissions between the models? There is almost a Given the authors they should be able to work this out for some of the model? Is it that they are using different emission inventories or making different choices about which species to include? Where are models with more or less chemical detail in this picture? Is there some rational for models that don't fit on the line? For example, there are two models which have almost the same EU AVOC emissions ∼22 Tg C a-1 but have very differing PAN responses. Can we understand these differences in terms of the model chemistry or meteorology? There is lots of interesting things that could be done here but the authors appear have performed a rather perfunctory analysis.

In conclusion I am rather conflicted about what to say. It looks like this paper has taken a long time to produce and a lot of work has gone into this paper probably over the last decade. I am however a bit confused by what has come out of this. I would suggest that the authors think about what the key points are, and then attempt to develop those further. There are plenty of people involved in this publication. It should be possible

to get something out of this. However, at the moment I don't feel the paper detailed or informative to make it suitable for publication.

---

## Referee Comment (RC2) · Anonymous Referee #2 · 30 Apr 2018

Fiore et al. very nicely highlights the importance of understanding and simulating PAN distributions to understand tropospheric ozone distributions. However, I was disappointed that there were not more specific results on the causes of model-measurement discrepancies. This is a very clearly written paper, though rather long relative to the new results presented. Previous work is referenced well. The figures clearly illustrate the points being made. One aspect of the paper that seems new to me is the use of the long-term mountaintop measurements for model evaluations and this is a nice presentation of their value.

I understand the interest and motivation to make use of the HTAP model simulations,

however, it seems to me there are a lot of limitations in using these simulations to understand PAN. The HTAP1 simulations did not use consistent emissions inventories across models, so it is very difficult to distinguish model chemistry and transport differences from purely emissions differences (in NOx and VOCs). The HTAP2 simulations, performed with more modern models, specified the emissions inventories to use for all simulations, and therefore might yield more conclusive results. However, in my experience, the simulation of PAN seems to be highly dependent on the BL dynamics of the model, and fine-scale chemistry, so it is difficult to see how much can be learned from monthly mean outputs, even with many models. It is my opinion that much more could be learned by the factors controlling PAN distributions using a single model with high time resolution output and comparison to the numerous aircraft measurements, as well as focused ground-based campaigns, that are available.

Previous studies have clearly illustrated that the chemical mechanism of a model has a big impact on PAN - not only the Emmerson and Evans studied referenced many times in this paper, but also Knote et al., Atmos. Environ., 2015. Previous work has also shown large multi-model differences, even when using the same emissions, in 3D models (e.g., Arnold et al., 2015; Emmons et al., 2015). So these points in this paper are not new.

Another concern I have is with the procedure for determining source attribution through emissions perturbations, which has been accepted by HTAP as standard procedure. The non-linearity of the chemistry in PAN and ozone formation will affect even relatively small perturbations such as 20% used here (see Butler et al., GMD discussions, 2018, and references therein). It seems to me that the large scatter shown in Figures 7 and 8 might largely be due to the non-linear chemistry in PAN formation on top of the differences in emissions and chemical mechanisms. Also, in Figs.7&8, what is the significance of the dashed line at 1.0 for the PAN ratio? Doesn't the r value correspond to a 1:1 line between y and x axes?

I am not entirely sure what to recommend for this paper. In its present state, it does

not seem to me to have enough new results to justify publication. Just as it is not really informative to evaluate ozone simulations without evaluating the precursors, perhaps more could be learned about the performance of the models if there were simultaneous evaluations of NOx and PAN-precursor VOCs to indicate why some models disagree so greatly with observations. I think the paper would also be strengthened by condensing the paper to focus on the really new results, and with less space used on the confirmation of previous findings.

---

## Author Comment (AC1) · 2 Jul 2018

**Responses to Referee Comments**

*The original referee comments are shown in italicized, black font.*
**Our replies are shown in bold, blue font.**

**We are very grateful to both reviewers for their thoughtful, constructively critical comments that we believe have enabled us to sharpen our focus and revise the manuscript to a stronger paper. In the process, we have revised every section of the manuscript. As our focus sharpened, we also revised the title to reflect better the key message of the manuscript: "Peroxy acetyl nitrate (PAN) measurements at northern mid-latitude mountain sites in April: A constraint on continental source-receptor relationships".**

**As detailed below, we now focus solely on April when our submitted manuscript pointed towards the largest potential for observations to constrain the inter-model range in PAN (and to some extent ozone) responses to anthropogenic emission changes within northern mid-latitude continents (source-receptor relationships, where the receptor is the mountaintop measurement site). We now clarify the novelty of this study: to exploit a multi-model ensemble in search of observational constraints (in our case, PAN measurements) to narrow the wide inter-model range in a quantity that is itself not directly observable (PAN source-receptor relationships, and ozone source-receptor relationships for cases where they correlate strongly with PAN). The revised manuscript now steps through a proof of concept demonstration of our approach to using PAN measurements, at one mountaintop site on each northern mid-latitude continent, to narrow the wide inter-model range in source-receptor relationships for PAN, and in some cases, ozone. During this revision we have conducted substantial new analyses, shown in Figures 4, 6a, and 8 in the revised version. Figure 4 demonstrates our "emergent constraint" approach, which we borrow from the climate science community, who first demonstrated the capacity for a model ensemble to identify observational constraints on unobservable quantities if those quantities strongly correlate with something observable. We have expanded other figures (Figures 3, 5 and 7 in the revised version) to include analyses of the HTAP1 models at Mount Waliguan in Asia, a location where short-term measurements have previously been made. Our new analysis supports this site – though other high-altitude sites will likely work as well -- as promising for long-term measurements to provide new constraints on source-receptor relationships, particularly for PAN originating in Asia. Finally, Figure 8 shows that, at least in April, PAN and ozone are correlated across the models even in surface air over the large continental-scale HTAP1 regions, thus indicating that our findings at individual mountaintop sites seem generalizable to broader scales.**

*Anonymous Referee #1*
*This paper attempts to understand the seasonal scale processes controlling the concentration of PAN at 5 mountain top sites in the northern mid-latitudes and to then use this as a tool to understand the processes controlling tropospheric O3. It attempts to do this through the framework of the HTAP1 model inter-comparison exercise, a Lagrangian modelling approach and the measurements made at the sites. The paper is relatively long and often feels a bit winding. It has a large number of authors and this committee approach is obvious in the paper.*

**As described above, we have thoroughly revised the paper, which is now shorter. Our major focus is to evaluate the potential for PAN measurements to provide constraints on the range of source-receptor relationships (i.e., in this case the PAN and ozone responses to 20% reductions in precursor emissions within a continental-scale HTAP1 region). Thanks to the reviewer for pointing out that the original version failed to articulate sufficiently clearly this main objective.**

*For all the effort that has gone into the paper my major concern is that it's not obvious what has been learnt and whether it is new and / or interesting? From the abstract the major conclusions seem to be: there are some differences between the attribution of the Eulerian and the Lagrangian approaches for a single site in Switzerland; VOCs are important for PAN production; ozone and PAN chemistries are qualitatively linked in models. I don't think any of these are particularly novel.*

**Thanks for this clear guidance. We have shortened discussion of these points to cite earlier work, allowing us to focus our message on the more novel aspects of identifying an "emergent constraint" on PAN and ozone source-receptor relationships, which requires a model ensemble for correlation analysis.**

*I think however, that with some extra work this paper could provide some interesting insights. These model simulations are old. They were run over a decade ago. How useful is this exercise if the models have now changed substantially? Has anything been learnt over the last decade which should be considered here. I don't think the authors can just ignore this issue.*

**Point well taken. We have worked both to include additional citations from the last decade, for example to the POLMIP multi-model analysis of airborne observations in the Arctic as suggested by Referee #2 below, and to better articulate the novelty of our analysis.**

**We have re-framed the manuscript and believe it now better communicates our original intention of demonstrating an approach to using a multi-model ensemble framework to assess and identify the potential for developing observational constraints for source-receptor relationships, which are notoriously difficult, if not impossible in some cases, to measure directly. We now refer to our analysis as a "proof of concept" because current measurements combined with the HTAP1 model ensemble are not sufficient to provide authoritative constraints due to the wide variability in measured PAN from year to year. As far as we are aware, the current generation of models continues to show large differences in ozone and PAN, implying that this analysis of a consistent set of simulations in a fairly large number of models to identify observational constraints should still be relevant.**

**In the final section (conclusions and recommendations), we suggest how future work with the next generation models – with additional model diagnostics archived (e.g., specific anthropogenic VOCs emitted and their treatment in chemical mechanisms) - could be probed alongside measurements for the same meteorological year(s) and thereby allow for clearer constraints.**

*One concern is that the paper investigates the importance of a number of issues in determining the processing controlling PAN for these mountaintop sites, but it makes little effort to then translate this understanding onto a larger spatial scale. How does PAN from EU VOCs make its way round the world? How strong is the PAN O3 relationship in different models globally? The paper discusses essentially the two sites that it has (one in the US and the multiple ones in almost exactly the same space in the Alps) and then seems to stop abruptly without widening its thinking to a more global or even hemispheric scale. What has been learnt which is more globally or regionally explicable? It would seem like a sensible next step to investigate the wider implications of the perturbation studies.*

**Thanks for this suggestion. We now show correlations between PAN and ozone over all nine source-receptor pairs in Figure 8. Importantly, we find the correlations across the models hold even in surface air, with significant correlations for similar source-receptor pairs as identified at the mountaintop sites, implying that these measurements can provide constraints that are relevant for larger scales.**

*I'm not sure of the usefulness of the Lagrangian approach in this paper. PAN's lifetime varies significantly with temperature, and whether it being produced or lost depending upon a complex set of*

*chemical reactions. The Langrangian approach may have some usefulness when looking at relatively short-lived, chemically simple tracers but it is not obvious that it has value when looking at PAN. The methodology is not described in any details and issues to do with subgrid processes (convection and boundary layer mixing) are not discussed at all. The authors need to show better its usefulness to PAN and then if they continue to use the measurements they should do something more useful with these calculatins. At the moment they show that the Lagrangian model gives different results than the Eulerian models and then appear to provide a very handwaving route to get to some form of 'consistency' between the different modelling framworks. At this point they essentially then proceed ignoring the issues raised by the Lagrangian method. The paper would come to the same conclusions without Langrangian section. I would therefore suggest it was removed.*

**We have followed this guidance and removed this section, with the exception of including the Lagrangian estimate for PAN at Jungfraujoch originating in the EU source region in the new Figure 4 (upper middle panel). For the reasons the referee outlines above, we have the most confidence in the estimate from the nearby EU source region. However, if the reviewer insists, we are open to removing even this small application of the trajectory-based analysis, as our conclusions are not based on the Lagrangian back-trajectory analysis.**

*The authors argue that there needs to be more measurements made from mountain top sites. I'm not convinced. Especially in the Alps there is a raft of data from these sites. It may be that making more mountain top measurements from other locations on other continents would be useful, but the authors provide little evidence for that. Where would they think that these measurements should be made? Himalayas, Urals, White Mountains, Rockies? Presumably from their model simulations they could make some suggests.*
**Thanks for this point. We believe our revised manuscript makes a stronger case for the utility of PAN measurements to select the subset of "best-performing" models and thereby narrow the range in model estimates of source-receptor relationships. We limited our analysis to sites where we knew PAN measurements had been made in the past, assuming that these sites would thus be the best candidates for future measurements. Although we did not use co-located measurements, there are obvious advantages in having other meteorological and chemical measurements available in contrast to a brand new location. Figure 2b shows that the inter-model range in PAN does not vary all that much over the northern hemisphere in April, suggesting that similar constraints to those identified here should be possible from various mountaintop locations.**

*Where there are interesting results (notably Figures 6 and 9) the authors don't really delve into the details. What explains the difference in AVOC emissions between the models? There is almost a Given the authors they should be able to work this out for some of the model? Is it that they are using different emission inventories or making different choices about which species to include? Where are models with more or less chemical detail in this picture? Is there some rational for models that don't fit on the line? For example, there are two models which have almost the same EU AVOC emissions _22 Tg C a-1 but have very differing PAN responses. Can we understand these differences in terms of the model chemistry or meteorology? There is lots of interesting things that could be done here but the authors appear have performed a rather perfunctory analysis.*
**While unfortunately our use of older simulations does hamper our ability to partition out the influence of different AVOC emission amounts versus the chemical mechanisms, we have nevertheless attempted to probe a little further the differences across models and assess why some of the models are falling off the regression line. We had already attempted to address this issue in the submitted manuscript in the figures that are now relegated to supplemental information (Supplemental Figures 2 and 3). We have recognized that the points we were trying to make were not as evident from those figures as we had intended and have thus developed new figures that we hope are easier to follow. Several of our revised figures now track, by assigning each model an**

**individual symbol (defined in Table 1), the placement of individual models. For the specific two models asked about by the reviewer, we can attribute those differences to model transport, and we now discuss this point in the text in Section 5:**

**"Differences in model transport (e.g., Arnold et al., 2015; Orbe et al., 2017) may also contribute to the inter-model differences in PAN source-receptor relationships, but our analysis of the HTAP1 idealized CO tracers reveals little correlation between inter-model differences in these idealized tracers (which have identical regional emissions and lifetimes applied in all of the models) and in the PAN source-receptor relationships sampled at these sites. Although we don't find any clear overall correlation, differences in the idealized CO tracers do help to explain some of the scatter in Figure 5; at Jungfraujoch for EU AVOC emissions of 22 Tg C a$^{-1}$, the lowest model (GISS-PUCCINI) has one of the lowest values of the COfromEU tracer, whereas the highest model (STOC-HadAM3) has the highest value of COfromEU."**

*In conclusion I am rather conflicted about what to say. It looks like this paper has taken a long time to produce and a lot of work has gone into this paper probably over the last decade. I am however a bit confused by what has come out of this. I would suggest that the authors think about what the key points are, and then attempt to develop those further. There are plenty of people involved in this publication. It should be possible to get something out of this. However, at the moment I don't feel the paper detailed or informative to make it suitable for publication.*

**We appreciate the opportunity to revise and have worked to deepen our analysis and better draw out the novel points in the revised manuscript.**

*Anonymous Referee #2*
*Fiore et al. very nicely highlights the importance of understanding and simulating PAN distributions to understand tropospheric ozone distributions. However, I was disappointed that there were not more specific results on the causes of model-measurement discrepancies. This is a very clearly written paper, though rather long relative to the new results presented. Previous work is referenced well. The figures clearly illustrate the points being made. One aspect of the paper that seems new to me is the use of the long-term mountaintop measurements for model evaluations and this is a nice presentation of their value.*

**Thanks for this comment, which has motivated us to re-frame the paper as described above. The new analysis in the revised version attempts to strengthen the case for the value of long-term mountaintop PAN measurements, at least in the month of April, for placing constraints on modeled source-receptor relationships for PAN, and in several cases ozone.**

*I understand the interest and motivation to make use of the HTAP model simulations, however, it seems to me there are a lot of limitations in using these simulations to understand PAN. The HTAP1 simulations did not use consistent emissions inventories across models, so it is very difficult to distinguish model chemistry and transport differences from purely emissions differences (in NOx and VOCs). The HTAP2 simulations, performed with more modern models, specified the emissions inventories to use for all simulations, and therefore might yield more conclusive results.*

**Agreed, and we have tried to articulate more clearly that our main purpose is to use this HTAP1 ensemble to demonstrate the value of the measurements, and to a lesser extent, to identify the factors contributing to inter-model differences. We have tried, in our revised Section 5, to improve the discussion of these factors beyond the submitted manuscript, including some discussion of the potential to provide a broad check on the amount of VOC emissions in this case where the models are using such a wide range for emissions (which we acknowledge in the text is of course convolved with differences in chemistry and transport):**

**"In light of the dependence of inter-model differences in PAN attributed to EU and NA during April and the corresponding regional AVOC emissions, we illustrate how one could extend our emergent constraints to the regional AVOC emission estimates shown in Figure 5. A major**

caveat underlying this analysis is the mis-match between meteorological years for the models and measurements as discussed above, and the underlying assumption that the relationships in Figure 5 can exclusively be attributed to differences in the AVOC emissions (as opposed to chemistry or transport). The observationally-constrained source-receptor relationships between PAN from NA and total PAN measured at Jungfraujoch and Mount Bachelor can be used to narrow the range of NA AVOC emissions narrows to 12-18 Tg C $a^{-1}$ (the low end is ruled out by the constraint imposed by PAN from NA at Jungfraujoch; the high end is ruled out by PAN from NA at Mount Bachelor). Similarly, the range for EU AVOC emissions would narrow to 16-25 Tg C $a^{-1}$."

In the same section, we also explore the role of model differences in transport, which is possible to study with the idealized regional CO tracers in which all models use the same emissions and apply the same lifetime.

*However, in my experience, the simulation of PAN seems to be highly dependent on the BL dynamics of the model, and fine-scale chemistry, so it is difficult to see how much can be learned from monthly mean outputs, even with many models. It is my opinion that much more could be learned by the factors controlling PAN distributions using a single model with high time resolution output and comparison to the numerous aircraft measurements, as well as focused ground-based campaigns, that are available.* We agree that for a deeper process-level analysis, a single model may be preferable if the ensemble has not archived the appropriate diagnostics. In our case, we do have information regarding differences in transport (which includes here the combined influence of boundary layer dynamics, convective mixing, as well as advection) from the idealized regional CO tracers. In Section 5 we have revised our discussion of these tracers:

"In contrast to AVOC, we find little relationship between the range in simulated PAN source-receptor relationships at the mountain sites and the model spread in regional anthropogenic $NO_x$ ($ANO_x$) emissions. Differences in model transport (e.g., Arnold et al., 2015; Orbe et al., 2017) may also contribute to the inter-model differences in PAN source-receptor relationships, but our analysis of the HTAP1 idealized CO tracers reveals little correlation between inter-model differences in these idealized tracers (which have identical regional emissions and lifetimes applied in all of the models) and in the PAN source-receptor relationships sampled at these sites. Although we do not find any clear overall correlation, differences in the idealized CO tracers do help to explain some of the scatter in Figure 5; at Jungfraujoch for EU AVOC emissions of 22 Tg C $a^{-1}$, the lowest model (GISS-PUCCINI) has one of the lowest values of the COfromEU tracer, whereas the highest model (STOC-HadAM3) has the highest value of COfromEU."

And:

"We consider next the importance that various models ascribe to a given source region relative to another source region. We first correlate the ratios of PAN from two different source regions with the total PAN simulated by the individual models in April. We find little relationship, with the exception of Mount Bachelor, where, intriguingly, the observational constraint implies that more PAN originating from EA should be present at Mount Bachelor than PAN originating from NA (Figure 6a). We interpret this as likely indicating that models with higher total PAN at Mount Bachelor are over-estimating North American influence at this mountain site sampling free tropospheric air. This interpretation is supported by the idealized CO tracer simulations (with identical regional emissions and the same lifetime applied in all the models), which suggest that some of the variance in the ratio of PAN from NA versus EA at Mount Bachelor is due to different transport from the two regions (Figure 6b)."

*Previous studies have clearly illustrated that the chemical mechanism of a model has a big impact on PAN - not only the Emmerson and Evans studied referenced many times in this paper, but also Knote et al., Atmos. Environ., 2015. Previous work has also shown large multi-model differences, even when*

*using the same emissions, in 3D models (e.g., Arnold et al., 2015; Emmons et al., 2015). So these points in this paper are not new.*

**Thank you. We now limit our discussion to citing these papers.**

**In the introduction:**
**"All of the HTAP1 models include PAN formation, but the chemical mechanisms and kinetic rate coefficients differ, with likely implications for long-range transport (Emmerson and Evans, 2009; Knote et al., 2015). A prior multi-model study found that even when models use the same emissions, PAN differs widely across the models, reflecting differences in simulated photochemistry (Emmons et al., 2015)."**

**In Section 5:**
**"Differences in model transport (e.g., Arnold et al., 2015; Orbe et al., 2017) may also contribute to the inter-model differences in PAN source-receptor relationships, but our analysis of the HTAP1 idealized CO tracers reveals little correlation between inter-model differences in these idealized tracers (which have identical regional emissions and lifetimes applied in all of the models) and in the PAN source-receptor relationships sampled at these sites. Although we do not find any clear overall correlation, differences in the idealized CO tracers do help to explain some of the scatter in Figure 5; at Jungfraujoch for EU AVOC emissions of 22 Tg C a$^{-1}$, the lowest model (GISS-PUCCINI) has one of the lowest values of the COfromEU tracer, whereas the highest model (STOC-HadAM3) has the highest value of COfromEU."**

**And in the conclusions:**
**"By focusing on April, our analysis largely minimizes complexities introduced by inter-model differences in biogenic, fire, and lightning sources that further complicate disentangling summertime discrepancies in simulated PAN and O$_3$ (e.g., Arnold et al., 2015; Emmons et al., 2015) and restricts inter-model differences to those associated with anthropogenic emissions and subsequent chemistry and transport. Nevertheless, we find a wide range in inter-model SRR relationships that reflects uncertainties in emissions and PAN yields from VOCs (Figure 5; see also Fischer et al., 2014; Arnold et al., 2015; Emmons et al., 2015). Different model representations of VOC chemistry also contribute to inter-model differences in PAN (e.g., Emmerson and Evans, 2009; Knote et al., 2015)."**

*Another concern I have is with the procedure for determining source attribution through emissions perturbations, which has been accepted by HTAP as standard procedure. The non-linearity of the chemistry in PAN and ozone formation will affect even relatively small perturbations such as 20% used here (see Butler et al., GMD discussions, 2018, and references therein). It seems to me that the large scatter shown in Figures 7 and 8 might largely be due to the non-linear chemistry in PAN formation on top of the differences in emissions and chemical mechanisms.*

**All of the models imposed 20% emission reductions, which should be sufficiently small as to avoid substantial deviations from linearity (see excerpt from main text below). None of the models use tagging schemes (the focus of the Butler et al., GMD discussions paper). We note that tagging schemes have their own limitations. For example, as discussed in *Jaffe et al., Elementa (accepted)*, "*tagging is more appropriate for source attribution than for estimating responses to emissions changes (e.g., Collet et al., 2014)*". The HTAP1 simulations are estimating responses to a 20% change in anthropogenic emissions within a region. It is unlikely that 20% perturbations applied in each of the models could induce non-linearities of such a large magnitude as to be a major factor in explaining the variations across the different models in Figures 7 and 8. In any case, we directly test the extent to which linearity holds with additional analysis in one of the HTAP models.**

**Please see the text at the end of Section 4:**

"**We note that for consistency with the modelled source-receptor relationships in Figure 4, which are the responses to 20% emission reductions in the source region, we divide the Pandey Deolal (2013) estimates for PAN originating in the EU by five to scale from their estimated "full contribution" (100%). This linear scaling of the PAN response between 20% and 100% may incur errors due to non-linear chemistry. We estimate this error to be ~10% by using an additional simulation with the FRSGCUCI model that sets European anthropogenic emissions of $NO_x$, CO and VOC to zero (a 100% perturbation; for intercontinental regions, this error reduces to <3%). Earlier work shows that the smaller non-linearity in PAN for intercontinental versus regional source-receptor pairs also holds for ozone (Fiore et al., 2009; Wu et al., 2009; Wild et al., 2012), and demonstrates approximate linearity between the simulated tropospheric ozone burden and ±50% of present-day global $NO_x$ emissions (Stevenson et al., 2006).**"

**Citations not in the main text:**

**Collet S, Minoura H, Kidokoro T, Sonoda Y, Kinugasa Y. et al. 2014. Future year ozone source attribution modeling studies for the eastern and western United States. _J Air Waste Manage Assoc_ 64(10): 1174–1185. DOI: 10.1080/10962247.2014.936629.**

**Jaffe, D.J., et al., Scientific assessment of background ozone over the U.S.: implications for air quality management, accepted at Elementa.**

_Also, in Figs.7&8, what is the significance of the dashed line at 1.0 for the PAN ratio? Doesn't the r value correspond to a 1:1 line between y and x axes?_
**We first note that these figures are now Supplemental Figures 2 and 3. The significance of the dashed line at 1.0 is to emphasize whether a model suggests one region or the other as the more important source of PAN at the mountaintop site. In the case of Mount Bachelor in Oregon, U.S.A., the revised manuscript now points out that the models suggesting higher NA over EA influence are probably less accurate and over-doing the transport from the NA region as indicated by the idealized regional CO tracers. We have added a note to the caption to explain this horizontal line at 1.0 for the PAN ratio in our new Figures 6a and 6b (in addition to Supplemental Figures 2 & 3) which illustrate this point:**
        **"The black dashed horizontal line at 1 separates the models suggesting a higher NA influence (above) versus higher EA influence (below)."**
**The captions on the supplemental Figures now include the following sentence:**
        **"The black horizontal dashed line at 1 separates the models suggesting one region versus the other as the larger influence."**

I am not entirely sure what to recommend for this paper. In its present state, it does not seem to me to have enough new results to justify publication. Just as it is not really informative to evaluate ozone simulations without evaluating the precursors, perhaps more could be learned about the performance of the models if there were simultaneous evaluations of NOx and PAN-precursor VOCs to indicate why some models disagree so greatly with observations. I think the paper would also be strengthened by condensing the paper to focus on the really new results, and with less space used on the confirmation of previous findings.

**We appreciate the opportunity to revise and have worked to deepen our analysis and better draw out the novel points in the revised manuscript.**

---

## Author Response (AR2)

**Responses to Referee Comments**

*The original referee comments are shown in italicized, black font.*
**Our replies are shown in bold, blue font.**

**We are very grateful to both reviewers for taking another look at our heavily revised manuscript. We respond below to additional, very helpful, suggestions from Referee #1.**

*Anonymous Referee #1*
*This paper has been significantly improved by the authors. It is shorter and more focused. The conclusions are supported by the work but it is still hampered by its use of models which are at least one generation behind the current state of the science. I don't see any obvious development in the intervening period that would completely invalidate the conclusions but it feels a little odd to be using results from these simulations.*
**Thank you for your guidance that helped us to focus and shorten the paper. We understand that these simulations are older than the reviewer would like. To our knowledge, this is the first application of the HTAP1 dataset to examine multi-model PAN source-receptor relationships, and this set of model simulations remains the most comprehensive available (i.e., a large set of sensitivity simulations performed consistently across 10+ models) for this type of analysis in search of observational constraints.**

*I would suggest that the conclusions are toned down a little bit. This feels like it is trying to prepare for grant funding to support PAN measurements into mountaintop sites. These observations are useful in the greater scheme of things but I'm not I would be as categorical about their value as the authors.*
**We have revised the conclusions and tried to "tone down" the recommendations for future measurements. We removed the final sentence of the conclusions, and moved the preceding sentences up to the second paragraph of the conclusions such that the paper now ends with the point about improving documentation of VOC emissions (including speciation) and their role as precursors to PAN and ozone in future multi-model efforts. We also deleted the sentences from the second paragraph that began to spell out "how-to" proceed with long-term PAN measurements. This second paragraph of the conclusions now reads:**
**"Establishing the strongest constraints possible on simulated SRRs for PAN and $O_3$ will require (1) measurements and simulations with chemical transport models that coincide, and (2) a sufficiently long measurement record to build a climatology suitable for evaluating chemistry-climate models that generate their own meteorology. Repeated sampling for the month of April may be sufficient to provide constraints on model responses to changes in anthropogenic emissions. PAN measurements over multiple seasons are necessary to evaluate model responses of PAN to climate change (e.g., by changing temperature and weather-sensitive precursor emissions) and the resulting influence on atmospheric $O_3$ and oxidizing capacity (e.g., Doherty et al., 2013). For example, changes in meteorology and biomass burning (Fischer et al., 2011; Zhu et al., 2015) such as those driven by ENSO (Koumoutsaris et al., 2008), as well as biogenic and lightning sources (Payne et al., 2017) vary from year to year and are expected to change as climate warms."**

*There are a number of typos in the paper which reflect the author's corrections but it would be useful if these were corrected before publication.*
**We hope we have now caught all the typos, but will continue to read carefully for typos at the proofs stage as well.**

*The figures are at rather low resolution. Figure 2 especially looks very fuzzy in the version I have.*
**The resolution looks much better in the original eps files that we will be uploading directly for the final publication, so we hope that these issues will be resolved.**

*Figure 3 would look better if the blue (multi-model mean) and the red (observations) were emphasised over the individual models. At the moment the most important symbols (blue and red) symbols are the least obvious.*

**We have worked to improve this figure. We now increase the symbol size and change to solid filled circles for the multi-model mean (blue) and the observations (red, except for year 2001 at Schauinsland which we show in blue to emphasize that this point is the best comparison point for the multi-model mean):**

[revised manuscript text omitted]